# Dietary restriction shapes intergenerational ribosome abundance and early growth of *Caenorhabditis elegans* offspring

Sigma Pradhan[1,2], Klement Stojanovski[1], Ferdinand Dellemann[1,2], Sacha Psalmon[1,2], Joel Tuomaala[1,2], Nicholas E. Stroustrup[3], Benjamin D. Towbin[1]*

1 Institute of Cell Biology, University of Bern, Bern, Switzerland, 2 Graduate School for Cellular and Biomedical Sciences, University of Bern, Bern, Switzerland, 3 Centre for Genomic Regulation, Barcelona, Spain

* benjamin.towbin@unibe.ch

## Abstract

Cells adjust their proteome to their environment. Most prominently, ribosome expression often scales near linearly with the cellular growth rate to provide sufficient translational capacity while preventing metabolically wasteful ribosomal excess. In microbes, such proteome adjustments can passively perpetuate through symmetric cell division. However, in animals, a passive propagation is hindered by the separation between soma and germline. This separation raises the question whether the proteome of animals is reset at every generation or can be propagated from parent to offspring across this barrier. We addressed this question for the intergenerational effects of dietary restriction in *Caenorhabditis elegans*, combining proteomics and live imaging. Under *ad libitum* feeding, the offspring of dietarily restricted mothers grew more slowly than progeny of well-fed mothers. However, this growth disparity was attenuated when mTORC1 signaling in the progeny was inhibited, creating conditions in which the protein-synthesis capacity at hatching is less limiting. Maternal inhibition of mTORC1 signaling, either ubiquitously or specifically in the pharynx, similarly reduced growth and ribosomal protein levels in offspring, whereas other growth-reducing perturbations, such as reduced insulin signaling or mTORC1 inhibition in the epidermis, did not reduce progeny ribosomal protein levels. We conclude that maternal physiology shapes ribosomal protein provisioning across the soma-germline boundary, thereby modulating early offspring growth in accordance with post-hatching ribosome demand.

## Introduction

Cells vary the allocation of their proteome resources to different tasks depending on their growth rate [1,2]. When growth is rapid, cells must highly express the protein

**Data availability statement:** The mass spectrometry proteomics data have been deposited to the ProteomeXchange Consortium via the PRIDE40 partner repository with the dataset identifier PXD060999. All other data are within the paper and its Supporting information files.

**Funding:** This work received funding from the Swiss National Science Foundation (SNSF) in the form of an Eccellenza Professorial Fellowship (PCEFP3_181204) to B.D.T., and project grants 320030L-227534 and 310030_219822, the Novartis Foundation for Medical-Biological Research (Grant #20A011), and the Berne University Research Foundation (34/22). B.D.T. is thankful to EMBO young investigator program (#5623) for support. N.E.S. received support from European Research Council under the European Union's Horizon 2020 Research and Innovation Programme (grant agreement no. 852201), The Spanish Ministry of Economy, Industry and Competitiveness to the EMBL partnership, the Centro de Excelencia Severo Ochoa (CEX2020-001049-S, MCIN/AEI/10.13039/501100011033), the CERCA Programme/Generalitat de Catalunya (NS),The Spanish Ministry of Economy, Industry and Competitiveness Excelencia awards PID2020-115189GB-I00 and PID2023-147692NB-I00. Some strains were provided by the CGC, which is funded by NIH Office of Research Infrastructure Programs (P40 OD010440). URLs of funders: https://www.snsf.ch, https://www.embo.org, https://www.stiftungmedbiol.novartis.com/, https://erc.europa.eu/, https://www.ciencia.gob.es/, https://cerca.cat/. The funders had no role in study design, data collection and analysis, decision to publish, or preparation of the manuscript. Si. Pr., K.S., F.D., Sa. Ps., J. T., and B. D. T. received salary support from Swiss National Science Foundation (SNSF) project and/or career funding.

**Competing interests:** The authors have declared that no competing interests exist.

**Abbreviations:** AID, auxin-induced degradation; AL, *ad libitum*; DR, dietary restriction; FDR, a false discovery rate; GO, Gene Ontology; IIS, insulin-like growth factor signaling; NGM, nematode growth medium; PDMS, polydimethylsiloxane; TMT, tandem-mass-tag.

synthesis machinery, including ribosomes, whereas maintaining such high ribosome expression in growth-limiting environments would be metabolically inefficient.

In bacteria, the cellular ribosome content follows a near-linear relation with the growth rate—a phenomenon termed the "growth law"—that optimizes growth across many conditions [3,4]. This relationship between growth and ribosome expression extends at least qualitatively to multicellular animals. At a cellular scale, ribosome expression varies by 3- to 10-fold among human tissues depending on their anabolic demands [5,6]. Similarly, at the organismal scale, ribosome expression often scales with the growth rate supported by the environment. For example, under conditions of dietary restriction (DR), where growth is slow, the nematode *Caenorhabditis elegans* reduces its ribosome expression accordingly [7].

Although the regulation of ribosomes in accordance with the growth rate is conserved between multicellular and unicellular forms of life, there is a fundamental difference in the propagation of ribosomes, and the proteome in general, across generations. In microbes, the proteome, and any adjustments to its content, can, in principle, be stably maintained across generations by the equal distribution of proteins between daughter cells during division. In multicellular animals, however, the separation of soma and germline, called the "Weismann barrier" [8], restricts the passive transmission of the proteome to subsequent generations.

Despite this barrier, the parental environment can have a pronounced impact on the physiology of the offspring [9], which in some cases can last over many generations [10]. Examples include maternal effects on resistance to pathogens [11], osmotic stress [12], temperature [13], or starvation [14].

While many studies have searched for and identified regulatory signals that sustain an altered gene expression profile in response to maternal environments, such as small RNAs, or histone and DNA modifications [10,15–17], a simpler yet crucial question has remained poorly explored: To what extent is the global partitioning of the proteome of newborn animals, and specifically its allocation to protein translation and ribosomes, determined by the environment and growth rate of their parents? Do animals, like microbes, quantitatively inherit their mother's proteome allocation to essential cellular machinery in accordance with their environment, or is the proteome globally reset between generations and independent of the parental environment?

We investigated these questions by examining the intergenerational effects of DR in *C. elegans*. In this model system, the proteomic and physiological response to diet within a generation is well established [7,18,19]. Similarly, physiological changes in the next generation have been extensively described, including effects on fertility, developmental robustness and plasticity, and shape and size of progeny [20–22].

We show that while DR profoundly alters the proteome within a generation, the proteome of the next generation is much less impacted by maternal diet. Ribosomal proteins represent a notable exception to this rule and maintain reduced levels also in offspring of dietarily restricted mothers. This reduced expression of ribosomal proteins had physiological consequences: when progeny of DR animals were exposed to *ad libitum* (AL) feeding, they displayed a delay in growth and development until they had synthesized normal ribosomal protein levels. This effect was

specifically dampened under conditions where the demand for protein synthesis in progeny was reduced. Ubiquitous and pharynx-specific maternal inhibition of mTORC1 signaling, but not other growth-inhibiting interventions, replicated the effect of DR. These data suggest that DR alters provisioning of ribosomal proteins to progeny via changes in maternal physiology that are not explained by a reduced maternal growth rate alone. Together, our study identifies intergenerational ribosome provisioning as a determinant of early progeny growth dynamics.

## Results

### Dietary restriction globally alters the proteome within a generation

To determine how dietary conditions alter the protein composition of *C. elegans* within and across generations, we employed tandem-mass-tag (TMT) proteomics. Animals were grown in liquid culture using *E. coli* HB101 as a food source. DR was imposed by a 5-fold food dilution compared to AL feeding ($2*10^8$ versus $10^9$ cfu/ml). Food dilution slowed down development to adulthood by 2 days per generation (changing the generation time from 3 days for AL to 5 days for DR-fed animals) (Figs 1A and S1A).

For both dietary conditions, we collected triplicate repeats at young adulthood stage (containing 1–3 eggs) (G0), and as L1 larvae of the next generation (G1) that were synchronized by overnight (16h) starvation. By TMT proteomics, we detected over 6,400 proteins across all replicates and conditions (6,454 in G0; 6,432 in G1). DR substantially altered protein expression in young adults with 104 proteins significantly downregulated, and 125 proteins significantly upregulated by more than 2-fold (Fig 1B, FDR < 0.05, S2A Table). Downregulated proteins were enriched for translational machinery, particularly in ribosomes, and for vitellogenins (yolk proteins) while upregulated proteins were enriched in plasma membrane and structural components of muscle cells (S1B Fig). These results are consistent with previous reports which measured DR-induced proteome changes in mutant backgrounds [7]. The data validate that, within a generation, DR induces substantial proteome-wide changes representing the reduced demand for protein translation under conditions of slower growth.

### Ribosomal proteins are downregulated intergenerationally after maternal DR

The effects of DR on the progeny were significantly weaker than those observed within a generation (Figs 1C and S1C, mean |$\log_2$(fold change)| = 0.3 (G0) versus 0.11 (G1); variance = 0.078 (G0), 0.016 (G1), $p = 3.5*10^{-126}$, Kolmogorov–Smirnov test) with only very few proteins significantly down- or upregulated by more than 2-fold (S1B Table). Overall, protein expression changes in progeny were not, or only very weakly, correlated to those in the parental generation (S1D Fig, $R^2 = 0.037$, $p = 1.74*10^{-46}$). These data suggest that proteome alterations after DR are not globally inherited across generations in *C. elegans*.

Although only three individual proteins passed stringent significance thresholds when analyzed individually (S1B Table), the maternal diet had a significant overall impact on the proteome composition, as evident from the separation along the first principal component (S1E and S1F Fig). This separation suggests that numerous proteins change consistently in expression in response to maternal diet, but since these changes are smaller than 2-fold, they may not be detectable at single protein resolution due to measurement noise.

To uncover intergenerational inheritance of the proteome composition beyond large (>2-fold) expression changes, we asked which Gene Ontology (GO) cellular component terms were enriched among proteins that were consistently up- or downregulated under DR compared to AL in both parents and progeny (|$z$-score| > 1 for AL-DR in each generation, Fig 1D). For this analysis, we did not apply the stringent significance thresholds used for identifying individually differentially regulated proteins, as our goal was to detect coordinated, modest effects among functionally related proteins. Nearly all significantly enriched GO terms among the downregulated proteins related to cytoplasmic or mitochondrial ribosomes. The only other GO terms enriched among consistently downregulated proteins referred to large subcellular compartments

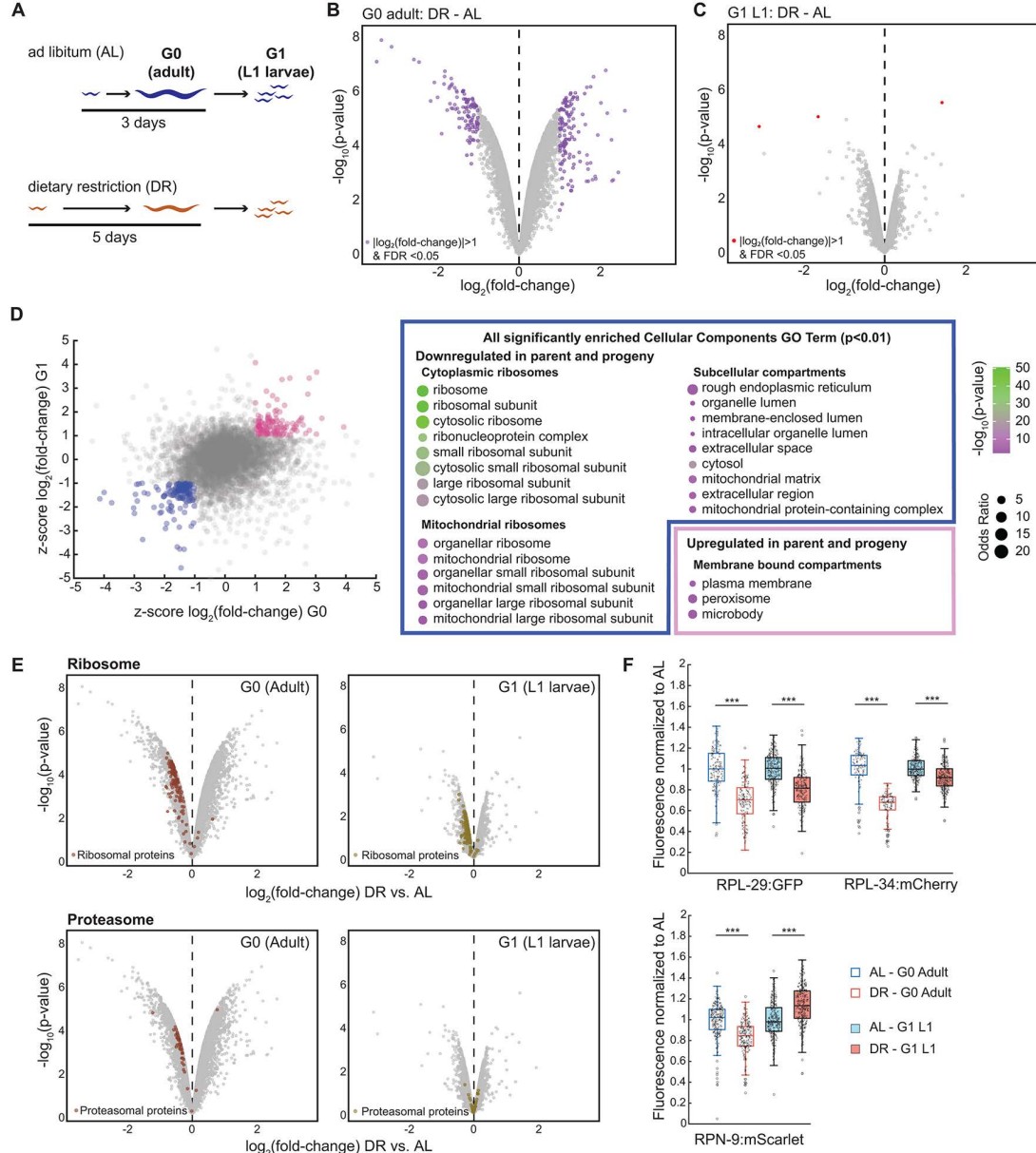

**Fig 1. Maternal dietary restriction (DR) reduces ribosomal protein expression in progeny. (A)** Experimental design of maternal DR. Animals were grown from L1 larvae to adulthood under *ad libitum* (AL) feeding (3 days) or DR (5 days). G0 Adults and their G1 progeny, synchronized as L1 larvae by overnight starvation, were collected for TMT proteomics in $n = 3$ biological repeats. **(B)** Volcano plot of relative changes in protein levels between DR and AL adults. Colored dots indicate proteins with a $|\log_2(\text{fold change})| > 1$ at a false discovery rate (FDR) < 0.05. **(C)** As (B), but for L1 larvae of G1 progeny. **(D)** Left: Scatter plot shows consistently upregulated (pink) or downregulated (blue) proteins after DR in parents and their progeny ($|z\text{-score}| > 1$). Right: All cellular component Gene Ontology terms enriched among consistently up- or down-regulated proteins with odds ratio > 1.5 and adjusted $p$-value < 0.01 (Benjamini–Hochberg correction). Circle size: odds ratio, Circle color: significance. GO enrichment is based on the $z$-scores trends marked in the scatter plot, without applying significance thresholds. **(E)** As (B) and (C), but with ribosomal and proteasomal proteins in color. **(F)** Quantification of fluorescence intensity (fluorescence per pixel) of indicated endogenously tagged proteins of parents exposed to DR or AL and their L1 progeny by live imaging after overnight starvation. RPL-29 and RPL-34 are ribosomal proteins, RPN-9 is a proteasomal protein. Data are normalized to the respective AL conditions. central line: median, box: interquartile range (IQR), whisker: ranges except extreme outliers (>1.5*IQR), individual values: crosses, extreme outliers: circles. For each condition, a total of at least $n > 110$ individuals (for G0) and at least $n > 230$ individuals (for L1 progeny) were measured over at least 3 days. *** indicate $p < 10^{-10}$ (Wilcoxon rank-sum test). See S2 Table for precise sample size and $p$-values, S1 Data.

with little functional specificity. Similarly, upregulated proteins were enriched in only a few GO terms, which related to membrane-bound compartments (Fig 1D, $p < 0.01$).

For both progeny and parent, nearly all detected ribosomal proteins were downregulated after DR (139/144 proteins for parents, 139/142 proteins for progeny, Fig 1E) and their median expression was significantly reduced ($-12\%$, $p = 1.04*10^{-22}$ for progeny and $-31\%$ for parents, $p = 4.8*10^{-23}$, Wilcoxon signed-rank test. S1G Fig). We did not observe such a dependence on maternal diet for other cytoplasmic proteins. For example, components of the proteasome were nearly unaffected by DR in the progeny (24/40 proteins positive fold-change, 16/40 negative fold-change, 1.5% mean change, $p > 0.05$, Wilcoxon signed-rank test) although these were strongly biased towards downregulation (41/44 proteins, $-23\%$ mean change, $p = 1.8*10^{-12,}$ Wilcoxon signed-rank test) in the parents (Figs 1E and S1G).

We validated these findings by live imaging using homozygous endogenously fluorescently tagged proteins, created by CRISPR/Cas9 mediated genome engineering, and functionally validated by growth assays and incorporation into polysomes (S10 Fig). When sampled by the same procedure as for mass spectrometry, the ribosomal proteins RPL-34: mCherry and RPL-29:GFP maintained reduced expression across generations, while the proteasomal protein RPN-9: mScarlet was downregulated in mothers but slightly upregulated in their progeny (Figs 1F and S1H).

Together, these data indicate that, although the proteome is globally altered under DR, the expression of most proteins shows little or no intergenerational change. However, unlike many other proteins, ribosomal proteins remain reduced in the next generation after maternal DR treatment.

We note that the quantitative change in ribosomal protein abundance was smaller in progeny than in their parents (Figs 1F and S1G). This difference nevertheless amounts to a significant change in the total amount of the progeny's protein mass allocated to ribosomes, given that ribosomal proteins comprise an estimated 10% of the worm's total protein mass [23].

## Reduced ribosomal protein levels after maternal DR correlates with slow growth and development

To investigate the physiological impact of maternal DR-induced ribosome reduction, we tracked individual progeny development using time-lapse microscopy. We loaded animals carrying an endogenous RPL-34:mCherry tag as embryos in agarose-based growth chambers with abundant food and monitored their development capturing images every 10 min until adulthood (Fig 2A). As previously described [24], this approach allowed us to derive precise volume growth rates, larval stage durations, and the volume at larval molts, which can be identified by a growth plateau before cuticular ecdysis [24].

Progeny of dietarily restricted mothers (called "DR progeny" from here onwards) had an 11% lower median RPL-34: mCherry intensity than progeny of *ad libitum* fed mothers (called "AL progeny") (Fig 2A and 2B). These measurements confirm the proteomics and imaging measurements of immobilized animals shown in Fig 1. Importantly, as these measurements in agarose chambers did not require synchronization by starvation at the L1 stage, these data also show that the reduction in RPL-34:mCherry occurs independently of larval arrest and immediately after hatching.

In addition to reduced RPL-34:mCherry concentration, DR progeny also had a 13% smaller volume (Fig 2C). Importantly, however, the observed 11% reduction in RPL-34:mCherry intensity (Fig 2A, 2B, and 2E) was not due to this size decrease, as we measured the change in mCherry concentration after size normalization. The total reduction in fluorescence without size normalization was 15%.

Maternal DR significantly impacted the speed of early development, delaying the L1 to L2 molt (M1) by an average of $2.7 \pm 0.2$ hours (s.e.m, $p = 1.5*10^{-39}$, Fig 2D). This delay results from a combination of the smaller initial hatching volume, and additionally the reduced volume specific growth rate (measured as $dV/dt/V = d\log(V)/dt$) that was correlated with a reduced RPL-34:mCherry expression (Fig 2E).

During development, the RPL-34:mCherry concentration progressively increased in DR progeny, reaching levels indistinguishable from that of AL progeny by M1 (Fig 2B and 2G). This gradual increase in RPL-34:mCherry levels was

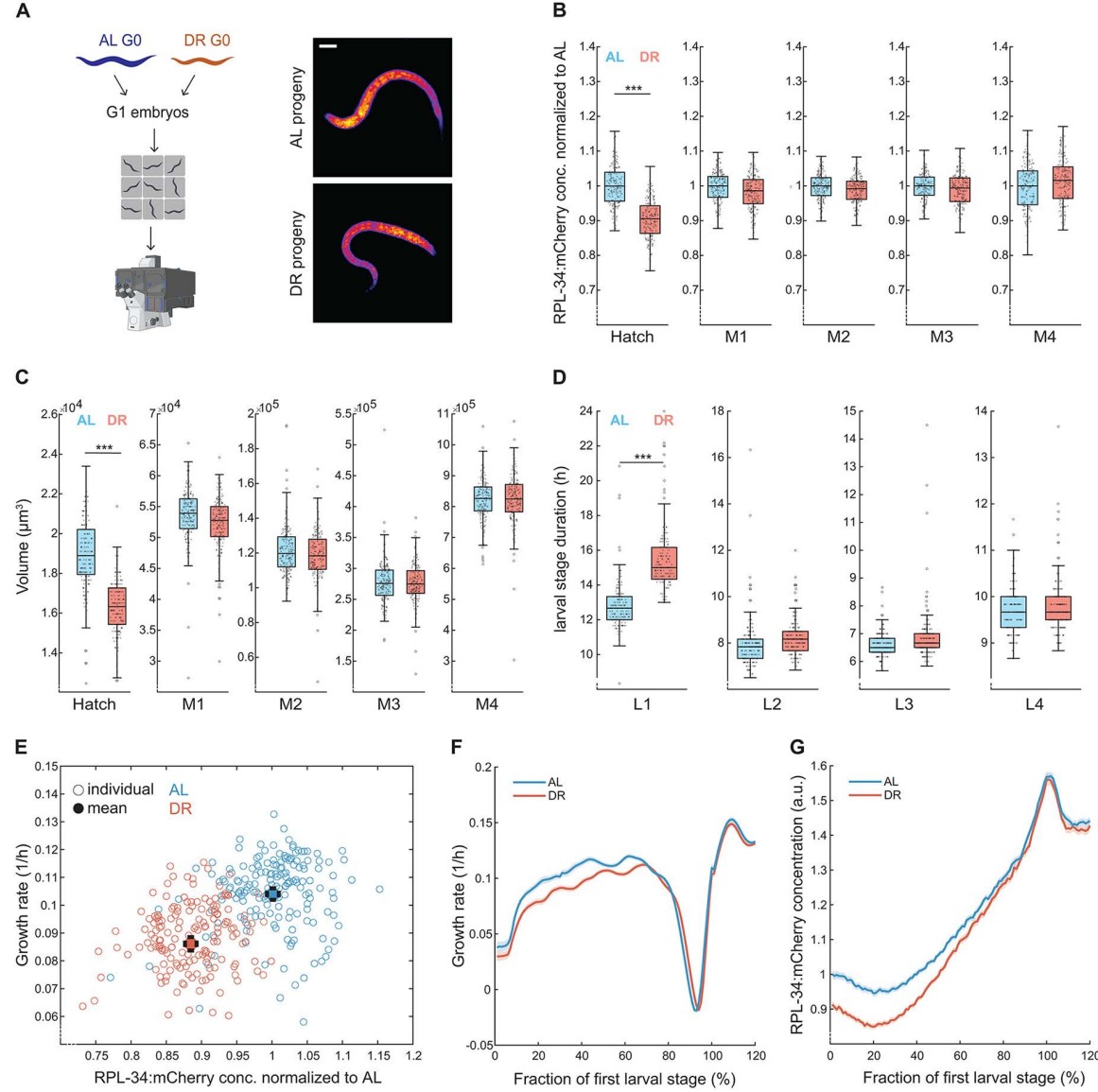

**Fig 2. Maternal DR reduces ribosomal protein and growth rate of progeny. (A)** left: Experimental design of growth assay. Embryonic progeny from AL and DR treated mothers were loaded into agarose chambers and monitored throughout their development by live imaging at 10-minute interval for measurement of larval stage durations, size, and RPL-34:mCherry fluorescence. Right: representative images (individuals closest to respective population median) of L1 larvae captured within 10 min after hatching in agarose chambers following DR or AL maternal diet. Contrast settings are equal for the two images. Yellow represents the high fluorescence intensity. Scale bar = 20μm. **(B)** RPL-34:mCherry concentration (fluorescence per pixel) of progeny of AL- (blue) and DR- (red) treated mothers after hatching and at molts M1 to M4. central line: median, box: interquartile range (IQR), whisker: ranges except extreme outliers (>1.5*IQR), individual values: crosses, extreme outliers: circles. For each condition, a total of at least $n \geq 169$ individuals were measured on at least 3 days. *** indicate $p < 10^{-10}$ (Wilcoxon rank-sum test). See S2 Table for precise sample size and p-values. **(C)** As **(B)**, but for volumes. **(D)** As **(B)**, but for larval stage durations. **(E)** Correlation between initial growth rate and RPL-34:mCherry intensity (fluorescence per pixel) after hatching. Individual measurements (circles) and population means (filled circles) shown for offspring of AL- (blue) and DR- (red) treated parents ($R^2 = 0.26$, $p = 6.69*10^{-24}$). Error bars: 2*s.e.m. Significance of difference between mean growth rates: $p = 2.4*10^{-26}$ (Wilcoxon rank-sum test) **(F)** Growth rate during L1 development. Individual trajectories were aligned at hatch point and M1 and re-scaled before averaging. Data between 100% and 120% represents the beginning of L2. Decline in growth rate represents the larval molt. Solid lines: mean, shaded regions: 95% confidence interval. Number of individuals and biological replicates as in **(B)**. **(G)** As **(F)**, but for RPL-34:mCherry intensity (fluorescence per pixel). See S2 Data. Fig 2A was created using BioRender. [Pradhan, S. (2026) https://BioRender.com/y76g797].

paralleled by a matching increase in the growth rate (Fig 2F), suggesting successful adaptation of DR progeny to AL conditions within the first larval stage. Indeed, beyond M1, development proceeded normally with no significant differences in subsequent larval stage durations, growth rate, ribosomal protein levels, or body volume between DR and AL progeny (Fig 2B–2D).

We conclude that the maternal diet impacts the initial ribosomal protein concentration and growth rate after hatching, and that animals restore normal growth rates over several hours of development and in coincidence with adjusting to normal ribosomal protein levels.

## Reduced ribosomal protein concentration after hatch causes slow growth

To determine whether reduced ribosomal protein levels were causally responsible for, or were merely correlated with slower growth, we developed a genetic system to quantitatively manipulate ribosomal protein abundance at hatch. We used an auxin-inducible degron (*aid*) and a GFP tag inserted at the endogenous locus of the *rps-26* ribosomal protein gene [25] and expressed the plant ubiquitin ligase Tir1 under the *sun-1* promoter, which is active in the germline and in the early embryo [26,27]. This approach allowed us to modulate ribosomal protein levels that L1 progeny have at hatch by titrating auxin (indole-3-acetic acid, IAA) concentration in the parental growth medium (Fig 3A) and use RPS-26:AID:GFP levels to measure the ribosomal protein concentrations.

We established auxin concentrations that, when applied to AL-fed mothers, reduced RPS-26:AID:GFP concentration by 12% at hatch (Fig 3B), close to the reduction observed after maternal DR (Figs 1F and 2B). Growth measurements in agarose chambers showed that this modest reduction in RPS-26:AID:GFP signal was indeed sufficient to recapitulate the slow growth observed after maternal DR (Fig 3C and 3D). Like progeny of DR animals, progeny depleted for RPS-26:AID:GFP by auxin-induced degradation (AID) at the time point of hatching similarly reached normal ribosomal protein levels during L1 development and recovered near normal growth rates (Fig 3D and 3E). We note that this recovery of ribosomal protein levels occurred slightly more rapidly following auxin-induced experimental depletion than after maternal DR (S2 Fig). This difference in the recovery dynamics indicates that while a reduction of ribosomal proteins is sufficient to cause a growth delay, other factors may additionally be involved with the developmental delay caused by maternal DR.

Maternal depletion of RPS-26:AID:GFP using ubiquitously expressed *eft-3p*:Tir1 yielded a similar intergenerational effect, although the effect on the growth rate was less pronounced (S3A–S3E Fig). This weaker effect is possibly due to partial germline silencing of Tir1 expression, as previously reported for *eft-3p* driven in transgenes [28]. Depletion of RPS-26:AID:GFP by Tir1 expressed in the distal germline from the *gld-1* promoter did not impact RPS-26:AID:GFP levels in progeny L1 larvae (S3F–S3J Fig). This dependence on the specific germline promoter used to express Tir1 may be due to differences in their expression strength, temporal dynamics, or spatial pattern within the germline.

## Maternal mTORC1/ RAGA-1 activity modulates progeny ribosomal protein provisioning

To explore whether maternal metabolic signaling can control ribosomal protein levels intergenerationally, we perturbed insulin-like growth factor signaling (IIS) and mTORC1 signaling, both of which have a conserved role in growth control [29–31].

Maternal effects of IIS on body size have been reported previously [14,19,22], and our experiments in agarose chambers confirmed these effects (S4A Fig). However, neither maternal somatic depletion of the IIS receptor DAF-2 via AID, nor mutation of its downstream effector *daf-16* altered progeny RPL-34:mCherry levels at hatching (S4 Fig). These data suggest that maternal IIS does not control ribosomal protein levels intergenerationally.

To perturb mTORC1 signaling, we maternally depleted RAGA-1, the sole *C. elegans* ortholog of the mTORC1 activators RagA/B [32–34], by AID using the ubiquitously active *eft-3p:*Tir1 transgene. As expected, RAGA-1 depletion slowed down organismal development (S5A Fig) and reduced the expression of RPL-34:mCherry (S5B Fig) in the maternal (auxin-treated) generation. Additionally, and contrasting IIS depletion, the maternal depletion of RAGA-1 also reduced RPL-34:mCherry concentrations in the progeny (Fig 4A). Maternal RAGA-1 AID also recapitulated several other maternal

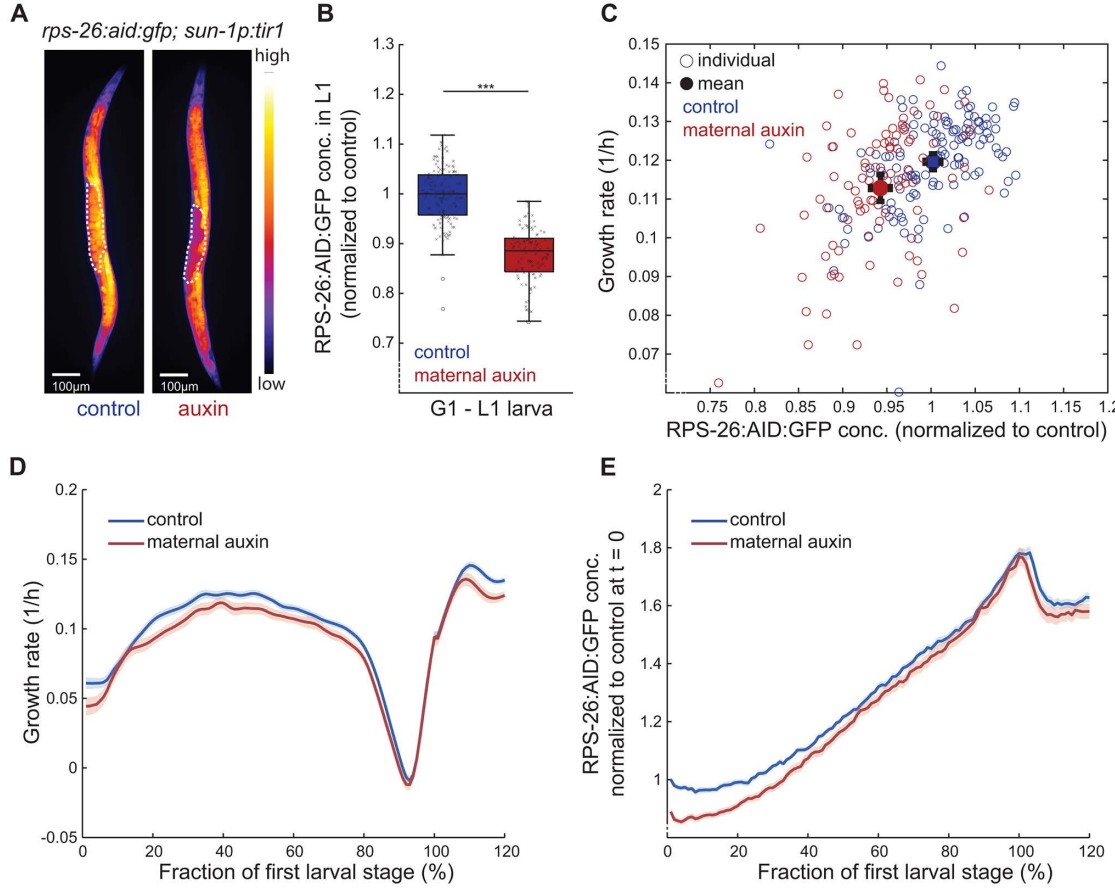

**Fig 3. Auxin-induced depletion of RPS-26:AID:GFP in germline and early embryos reduces progeny growth and ribosomal protein levels. (A)** Fluorescence microscopy image of *C. elegans* with endogenously inserted *rps-26:aid:gfp* tag expressing *sun-1p:*Tir1 treated with 250 μM auxin (right) or control (left). White dotted line: embryos in uterus. Scale bar: 100 μm. **(B)** Quantification of RPS-26:AID:GFP fluorescence (intensity per pixel) in newly hatched L1 larvae from control and auxin-treated mothers. central line: median, box: interquartile range (IQR), whisker: ranges except extreme outliers (>1.5*IQR), individual values: crosses, extreme outliers: circles. Total number of individuals $n = 131$ and 100 for control and auxin-treated, number of experiments = 3, *** $p < 10^{-33}$ (Wilcoxon rank-sum test). **(C)** Correlation between initial growth rates and RPS-26:AID:GFP concentrations after hatching. Individual measurements (circles) and population means (filled circles) shown for offspring of control (blue) and auxin-treated (red) mothers ($R^2 = 0.25$, $p = 1.3*10^{-15}$). Error bars: 2*s.e.m. Significance of difference between mean growth rates: $p = 1.8*10^{-3}$ (Wilcoxon rank-sum test). **(D)** Growth rate during L1 development. Individual trajectories were aligned at hatch point and M1 and re-scaled before averaging. Range between 100% and 120% represents the beginning of L2. Solid lines: mean, shaded regions: 95% confidence interval. Number of individuals and biological replicates as in (B). **(E)** As (D), but for RPS-26:AID:GFP concentration (fluorescence per pixel). See S3 Data.

DR-associated characteristics, slowing down development during the L1 stage, reducing the size of L1 larvae at hatching (Fig 4A), and slowing down the growth rate after hatching in correlation with reduced RPL-34:mCherry levels (Fig 4B). As for maternal DR, these phenotypic changes converged towards those of untreated animals within the first larval stage (Fig 4B), such that animals experiencing maternal RAGA-1 depletion were nearly indistinguishable from untreated controls from the L2 stage onwards (S6A–S6C Fig).

In summary, while altered maternal IIS did not impact progeny ribosomal protein levels, maternal RAGA-1 AID recapitulated the reduction of ribosomal proteins in progeny and other phenotypes caused by maternal DR. Notably, despite this strong correlation between RAGA-1 AID and DR, we do not exclude that the intergenerational effects of DR involve additional pathways, or that DR can act through mTORC1-independent, redundant mechanisms. Since RAGA-1 AID does

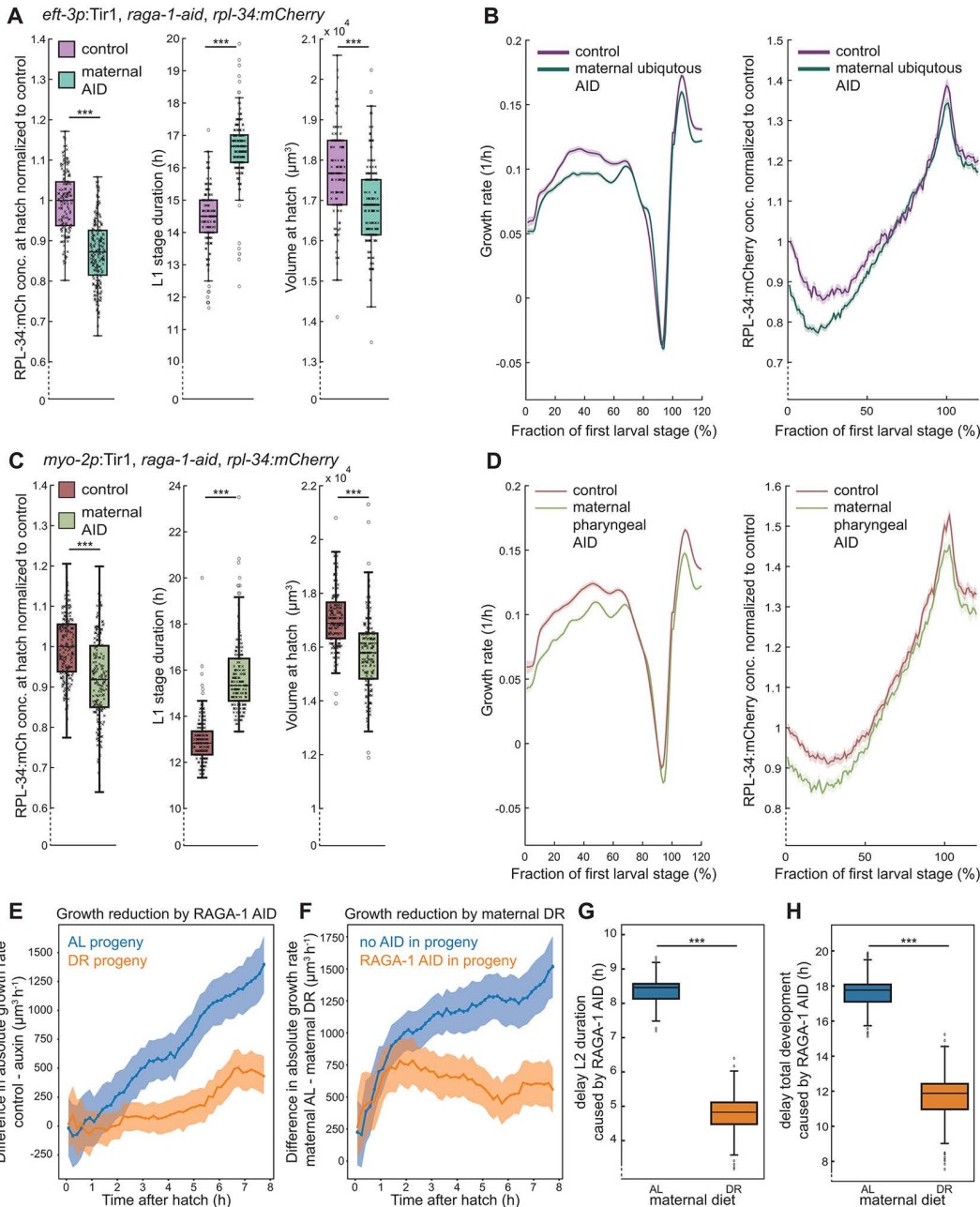

**Fig 4. Maternal depletion of RAGA-1 phenocopies maternal dietary restriction. (A)** RPL-34:mCherry concentration (fluorescence per pixel, normalized to control), L1 larval stage duration, and volume at hatch after maternal RAGA-1 depletion (green, 500 μM maternal auxin) and control (magenta, no maternal auxin). No auxin was added to progeny generation. central line: median, box: interquartile range (IQR), whisker: ranges except extreme outliers (>1.5*IQR), individual values: crosses, extreme outliers: circles. $n \geq 150$ from 2 days **(B)** Growth rate and RPL-34:mCherry concentration of progeny with and without maternal RAGA-1 depletion. Individual trajectories were aligned at hatch point and M1, and re-scaled before averaging. Range between 100% and 120% represents the beginning of L2. Solid lines: mean, shaded regions: 95% confidence interval. **(C, D)** As (A, B), but for pharynx-specific maternal RAGA-1 depletion. Strains expressing *myo-2p*:Tir1 (green) and not expressing Tir1 (red) were both treated maternally with 500 μM auxin. No auxin was added to progeny generation. $n \geq 202$ from 3 days **(E)** Progeny of AL fed (blue) and DR mothers (orange) were grown with or without ubiquitous RAGA-1 AID (*eft-3p*:Tir1) in agarose chambers with abundant food. The difference in the absolute growth rate (volume / time) between progeny exposed to 500 μM auxin or to vehicle control is plotted as a function of time after hatching. Shaded area: 95% CI. $n > 49$ individuals from 3 different days. **(F)** As **(E)**, but for the growth rate disparity between DR and AL progeny under RAGA-1 AID and control conditions. **(G, H)** As (E), but for the difference in L2 duration and total developmental duration from hatch to M4 (right). *** indicate $p < 10^{-10}$ (Wilcoxon rank-sum test). See S2 Table for precise sample size and *p*-values, S4 Data.

not represent a complete loss of mTORC1 activity, classic epistasis analysis is not suited to resolve strict pathway dependence [35].

### Tissue-specific depletion of RAGA-1 in the pharynx, but not in the epidermis reduces progeny ribosomal proteins

To determine whether the observed reduction of progeny ribosomal protein depends on somatic versus germline or embryonic RAGA-1 activity, we depleted RAGA-1 by AID specifically in the pharynx or in the epidermis (using *myo-2p:*Tir1 and *col-10p:*Tir1). We previously reported that RAGA-1 AID in either of these tissues slowed down growth, while keeping pharynx-to-body proportions nearly unchanged [34].

Maternal depletion of RAGA-1 in the pharynx significantly reduced RPL-34:mCherry levels and the hatching volume of the progeny, and delayed larval development, similar to ubiquitous maternal RAGA-1 AID and maternal DR (Figs 4C, 4D, and S6D–S6F). Epidermal depletion of RAGA-1 did not reduce progeny ribosomal proteins in the next generation (S7 Fig), despite strongly affecting maternal growth [34]. Together, these data show that depletion of RAGA-1 in specific maternal somatic tissues is sufficient to alter progeny ribosomal protein levels. Yet, this intergenerational effect depends on the tissue that is affected, and slower maternal growth alone is not sufficient to reduce progeny ribosomal proteins.

### Maternal RAGA-1 AID alters ribosomal protein provisioning in the maternal germline, but not their synthesis in the embryo

To distinguish whether maternal RAGA-1 depletion impacts ribosomal protein synthesis in the progeny or maternal ribosomal protein provisioning, we measured RPL-34:mCherry levels in developing embryos. Both maternal depletion of RAGA-1 in the pharynx or ubiquitous maternal RAGA-1 depletion reduced ribosomal protein levels already in embryos without changing the rate of change during later embryo development (S8A and S8B Fig). These data suggest that maternal RAGA-1 AID does not impact the synthesis of ribosomal proteins during embryogenesis of the progeny, but their initial loading.

Consistently, RPL-34:mCherry was significantly reduced in the germline of mothers ubiquitously depleted for RAGA-1 by AID (S8C, S8D, S8G, S8H Fig). This effect was increasingly pronounced towards the proximal end of the germline (S8E Fig), and anti-correlated with the increasing size of the developing oocytes located in this region (S8F Fig). These data suggest that the balance between ribosomal protein synthesis and oocyte growth during late oogenesis influences the ribosomal protein concentration in the progeny, although we cannot entirely exclude an additional contribution of ribosomal protein synthesis in other regions of the germline as well.

### Relative developmental delay after maternal DR is less pronounced when progeny growth is slow

To investigate the physiological significance of intergenerational ribosomal protein control, we asked how the growth impairment of the offspring by maternal DR depends on the growth conditions of the progeny. Specifically, we predicted that if ribosomes of DR progeny were limiting for their growth under AL conditions, then the growth disparity caused by maternal diet should be less pronounced when progeny were exposed to conditions with a low demand for protein synthesis capacity.

To test this, we exposed mothers to AL or DR conditions and compared their progeny subjected to ubiquitous mTORC1 depletion via RAGA-1 AID with progeny grown under control conditions. RAGA-1 AID experimentally mimics the reduced demand for ribosomes in progeny (S5 Fig) that would be expected under continued DR, while (unlike DR) being experimentally feasible in agarose growth chambers.

Consistent with our prediction, the reduction in growth rate caused by RAGA-1 AID was less pronounced for DR progeny than for AL progeny during the first few hours after hatching (Figs 4E and S9A). Conversely, the growth disadvantage of DR progeny was less pronounced when progeny was exposed to RAGA-1 AID (Figs 4F and S9B). These effects are consistent with a model, where DR progeny has the strongest delay under conditions that allow rapid growth and

where the deviation from optimal ribosome levels is largest. Similarly, the delay in total developmental duration caused by RAGA-1 AID was less pronounced for DR progeny compared to AL progeny, with the strongest effect occurring in the L2 stage (Figs 4G, 4H, and S9C–S9D).

Together, these data suggest that reduced maternal ribosome provisioning has a larger growth cost when the conditions of progeny allow for rapid growth.

## Discussion

Organisms adjust their proteome to their environmental conditions. Previous work has systematically analyzed this response in unicellular microbes and revealed that many proteins scale near-linearly with the growth rate [1,2]. A similar relation occurs in animal cells for ribosomes, which scale with the growth rate and the anabolic demand of cells [7]. Here, we used DR of *C. elegans* to ask if the proteome composition is propagated across multiple generations in a multicellular animal or is reset at every generation.

A key finding of this study is that most diet-induced changes to the proteome are not propagated across generations. However, ribosomal proteins are an exception to this rule and are downregulated in the dietarily restricted parental generation, as well as in their progeny (Fig 1). Whereas previous work has documented intergenerational information transfer, e.g., through chromatin modifications or small RNAs [10,36,37], our findings reveal that also the transfer of the core metabolic machinery can influence offspring phenotypes.

Most ribosomes that *C. elegans* expresses when hatching are maternally provided, and maternally provided ribosomes are sufficient to complete embryogenesis [38]. Consistently, we found no evidence for differential ribosomal protein production during embryogenesis in response to maternal mTORC1 signaling (S8 Fig). Instead, our data suggest that maternal conditions impact the deposition of ribosomes into the progeny cytoplasm, although we cannot entirely exclude an additional contribution of other mechanisms.

Our AID experiments show that maternal mTORC1 signaling modulates progeny ribosomal protein content in a tissue-dependent manner. Pharyngeal RAGA-1 depletion reduced ribosomal protein levels in the offspring, whereas epidermal depletion did not, despite both perturbations impairing maternal growth [34]. These findings indicate that the progeny ribosomal protein content does not simply track the maternal growth rate, but instead additionally reflects other changes in maternal physiology transmitted across the soma-germline boundary.

One possible mechanism for this cross-tissue effect is that reduced RAGA-1 activity in the pharynx decreases food intake and thereby limits nutrient supply to the germline. The germline may in turn respond to this nutrient shortage by reducing the allocation of resources to ribosome synthesis for embryonic deposition. Notably, we previously showed that epidermal and pharyngeal RAGA-1 depletion both elicit a systemic growth response that preserves pharynx-to-body size proportions [34], making pharyngeal size alone unlikely to account for the observed effects. While our experiments demonstrate that the germline adjusts its regulatory state in response to reduced mTORC1 activity in a specific tissue, future work will be required to disentangle the relative contributions of nutrient-based cues and canonical signaling pathways to this germline-soma communication.

Other mechanisms, such as differential yolk loading [22,39] and the regulation of ribosome composition and modification [40,41], have been related to intergenerational effects. While these processes likely also contribute to growth control under DR, our observation that quantitatively matched, direct degradation of a ribosomal protein had a similar effect on early progeny growth rates as DR (Figs 2 and 3) support a causal role for maternal ribosome deposition.

What could be the physiological impact of intergenerational ribosome control? One possibility is that reduced allocation of resources to ribosomal protein production and growth could free resources for other purposes, such as stress resistance or survival under complete starvation. Consistently, progeny of dietarily restricted mothers exhibit improved developmental robustness during prolonged starvation [22], although it remains unclear whether these benefits arise directly from altered ribosome expression. A second, non-exclusive possibility is that lowering embryonic ribosome deposition reduces

the metabolic cost of reproduction for the mother, while having a lesser effect on progeny growth under continued DR. While sustained DR of progeny cannot be implemented in agarose chambers, we measured the growth reduction caused by maternal DR under RAGA-1 AID, which similarly reduces the ribosomal protein demand for maximal growth. Consistent with the model above, growth disparities caused by maternal diet were less pronounced when progeny were exposed to RAGA-1 AID (Fig 4E–4H).

In conclusion, our work provides a systematic analysis of intergenerational proteome regulation in response to maternal diet. It highlights the allocation of proteome resources to ribosomes as a mechanism of intergenerational plasticity that modulates early progeny growth as a function of the post-hatching demand for protein synthesis capacity.

## Materials and methods

### *C. elegans* strains and maintenance

Animals were grown and maintained at 25 °C using OP50-1 *E. coli* on nematode growth medium (NGM) according to standard procedures. Fluorescently tagged proteins were all endogenously modified by CRISPR/Cas9 and tags were kept homozygous in all experiments. We confirmed that fluorescent ribosomal protein tags did not slow down development, although a slight decrease in body size was detected. All ribosomal proteins efficiently incorporated into polysomes, confirming the functionality of the tagged proteins (S10 Fig). All experiments were conducted using matched tagged strains as controls.

The following strains were used in this study

N2: Bristol N2 wild isolate: used in Figs 1A–1E and S1A–S1E

PHX1880: *rpl-34(syb1880) [rpl-34:RPL-34:mCherry] IV* used in Figs 1E, S1F, 2, S2, S4D, S4E, S10A, S10B, and S10E

PHX1945: *rpl-29(syb1945) [rpl-29:GFP]* used in Figs 1F, S1F, S10A, S10B, and S10C

CER620: *ubh-4(cer68[ubh-4::eGFP]) rpn-9(cer203[rpn-9::wrmScarlet]) II* used in Figs 1F and S1F, previously published in [42]

wBT340: *rubSi4[rps-26::aid::gfp] I; unc-119(ed3) III; ieSi38 [sun-1p::TIR1::mRuby::sun-1 3′UTR + Cbr-unc-119(+)] IV* used in Figs 3A–3E and S2

SUR20: *rubSi4[rps-26::aid::gfp] I; ieSi57[eft-3p::TIR1::mRuby::unc-54 3′UTR + Cbr-unc-119(+)] II* used in S3A–S3E Fig, previously published in [25]

SUR21: *rubSi4[rps-26::aid::gfp] I; [Pgld-1::TIR1::mRuby::gld-1 3′UTR] II* used in S3F–S3J Fig, previously published in [25]

wBT440: *ieSi57 II; daf-2(bch40) III; rpl-34(syb1880) [rpl-34:RPL-34:mCherry] IV:5.83* used in S4A–S4C Fig

wBT371: *daf-16(mu86)I; rpl-34(syb1880) [rpl-34:RPL-34:mCherry] IV:5.83* used in S4D and S4E Fig

wBT316: *raga-1(wbm40) [raga-1::AID::EmGFP] II; xeSi376[Peft-3::TIR1::mRuby::unc-54 3′UTR, cb-unc-119(+)] III; rpl-34(syb1880) [rpl-34:RPL-34:mCherry]* used in Figs 4A, 4B, 4E–4H, S5, S6A, S6B, S6C, S8A, S8C–S8F, and S9

wBT373: *reSi1 [col-10p::TIR1::F2A::mTagBFP2::NLS::AID::tbb-2 3′UTR] (I:-5.32), raga-1(wbm40) [raga-1::AID::EmGFP*] II; rpl-34(syb1880) [rpl-34:RPL-34:mCherry] IV:5.83* used in S7 Fig

wBT374: *raga-1(wbm40) [raga-1::AID::EmGFP*] II; rpl-34(syb1880) [rpl-34:RPL-34:mCherry] IV:5.83* used in Figs 4C,4D, S6D, S6E, S6F, and S7

wBT379: *wbm40 [raga-1::AID::EmGFP] ieSi60[myo-2p::TIR1::mRuby::unc-54 3′UTR+Cbr.unc-119(+)] II; (?unc-119(ed3)? III); rpl-34(syb1880) [rpl-34:RPL-34:mCherry] IV:5.83* used in Figs 4C, 4D, S6D, S6E, S6F, and S8B

wBT553: *rubSi4[rps-26::aid::gfp] I* used in S10A, S10B, and S10D Fig

HW1939: *xeSi296[eft-3p::luc-gfp::unc-54 3′UTR, unc-119(+)] II.* used in S10A and S10B Fig, previously published in [43]

## Liquid culture growth medium for dietary restriction

DR was attained by growth in liquid culture based on previous protocols [44]. For liquid culture, the bacterial strain *Escherichia coli* HB101 was diluted in S-basal (supplemented with 10mM Potassium citrate, 3mM $CaCl_2$, 3mM $MgSO_4$, trace metal solution, 5mg/l cholesterol, 100μg/l carbenicillin, 10μg/l Nystatin) as a food source at $2*10^8$ cfu/ml for DR conditions. AL conditions were identical, except with $10^9$ cfu/ml. Prior to liquid culture, animals were grown on NGM plates at 25 °C for at least three generations. To obtain synchronized populations, gravid adults were bleached, and their embryos were allowed to hatch overnight in S-basal buffer without ethanol or cholesterol. These synchronized L1 larvae were then cultured in liquid culture medium at a density of 100 worms/ml in 50-ml falcon tubes rotated at 25 °C. At this worm density, no change in food concentration due to consumption was detectable during the course of our experiment.

## Mass spectrometry sample preparation

For mass spectrometry of adults, animals subjected to AL feeding or DR were collected after cultivation in liquid for 3 or 5 days, respectively, starting from synchronized L1 larvae. For G1 progeny, eggs were released from gravid adults cultured under DR or AL by bleaching and remnants of incompletely bleached adults were removed by filtering through a 40 μm mesh (Pluriselect). L1s were allowed to hatch overnight in S-Basal (without cholesterol or ethanol) on an orbital rotor and unhatched eggs were removed by filtering through a 20 μm mesh (Pluriselect) to obtain synchronized, arrested L1 larvae. For each condition, three biological replicates were obtained. Number of animals per repeat: 5,000, 10,000, and 100,000 animals each for AL adults, DR adults, and L1 progeny, yielding 30–50 μg protein for each sample. To extract protein, worm pellets were resuspended in 100 μL of lysis buffer (PBS pH7.4 containing 0.025% Triton X-100 and protease inhibitors (Sigma P8340) and flash frozen in liquid nitrogen followed by thawing at 37 °C three times and subsequently sonicated for 15min using 30-s on/off cycles in a sonicator bath (Bioruptor). Insoluble protein was removed by centrifugation and the concentration of soluble protein was determined by a micro-BCA assay (Thermo Fischer Scientific). G0 and G1 samples were each quantified in a separate TMT 6-plex experiment.

## TMT mass spectrometry and analysis

The raw output files of FragPipe (protein.tsv files) were processed using the R programming language (ISBN 3-900051-07-0). Contaminants and reverse proteins were filtered out and only proteins that were quantified with at least 2 razor peptides (Razor.Peptides ≥ 2) were considered for the analysis. 7475 proteins passed the quality control filters. $Log_2$-transformed raw TMT reporter ion intensities ('channel' columns) were first cleaned for batch effects using the 'removeBatchEffect' function of the limma package [45] and further normalized using the 'normalizeVSN' function of the limma package (VSN—variance stabilization normalization [46]). Proteins were tested for differential expression using a moderated t *test* by applying the limma package ('lmFit' and 'eBayes' functions). The replicate information was added as a factor in the design matrix given as an argument to the 'lmFit' function of limma. A protein was annotated as a hit with a false discovery rate (FDR) smaller 0.05 and an absolute fold-change of greater 2. The mass spectrometry proteomics data have been deposited to the ProteomeXchange Consortium via the PRIDE [47] partner repository with the dataset identifier PXD060999. GO enrichment analysis was conducted in R using clusterProfiler (Bioconductor) with the org.Ce.e.g.,db annotation package. Enrichment was calculated via the hypergeometric test (enrichGO()), applying Benjamini–Hochberg correction. The background universe consisted of all quantified genes in the dataset. Only GO terms from the Cellular Component ontology with FDR < 0.01 and odds ratio > 1.5 were considered significant.

## Single time point imaging

*C. elegans* strains with endogenously tagged ribosomal or proteasomal proteins, collected at early adulthood stage or as overnight hatch-synchronized larvae were imaged at a single time point. Worms were mounted onto 2% agarose pads on slides and a drop of 10 mM levamisole was used to anesthetize the worms. Single time point imaging in Fig 1 was conducted on Nikon Ti2 epifluorescence microscope using a 10× air objective (NA = 0.45) for adults and using a 20× air objective (NA = 0.75) for L1 larvae. Images for S8C–S8H Fig were acquired on a dual camera spinning disk confocal microscope (Nikon Ti2, Yokogawa W1) using a 40×/ 1.3 NA objective and 2 × 2 binning on a Photometrics Kinetix camera.

## Live imaging in microchambers

For live imaging of progeny of animals cultured in liquid by DR or AL, adults were collected from liquid culture filtered through a 40 µm mesh cell strainer (Pluriselect) to remove younger animals. Egg laying was then induced by incubating adults in 35 mM serotonin in S-basal on an orbital rotor at 25 °C for 1 hour and the released embryos were used for loading into agarose-based growth chambers. For experiments using maternal auxin treatment (Figs 3, 4, S3, S4, S5A–S5D, and S6), the parental generation was cultivated on standard NGM plates supplemented with auxin, and eggs were picked directly from these plates for loading into agarose chambers.

Arrayed agarose microchambers were manufactured using 4.5% agarose dissolved in S-basal (containing 5 µg/ml cholesterol). An inverse replicate was created from a polydimethylsiloxane (PDMS) stamp, as described by Turek and colleagues [48] with minor modifications described by Stojanovski and colleagues [24]. Chamber dimensions were 600 µm × 600 µm × 20 µm. For imaging larval development, chambers were filled with the bacterial strain OP50–1 scraped from a standard NGM plate as a food source, 1.5- to 2-fold stage embryos were manually placed in individual chambers using an eyelash pick, and agarose chambers were sealed by inverting onto a 3.5 cm wide dish with a high optical quality gas-permeable polymer bottom (ibidi). 3% low melting temperature agarose dissolved in S-basal (containing 5 µg/ml cholesterol) was pipetted around the agarose chamber array and topped with ~300–500 µl PDMS to prevent evaporation. Finally, the dish was sealed with parafilm and mounted on a custom-made sample holder for microscopy.

All time-lapse imaging was conducted on a Nikon Ti2 epifluorescence microscope equipped with a Hamamatsu Flash 4 sCMOS camera using a 10× air objective (NA = 0.45). Software-based autofocus was performed at each time point using the default parameters in Nikon NIS software. GFP excitation was at 470 nm, and mCherry excitation was at 575 nm, with 10 ms exposure times using a SpectraX light source (Lumencor). Images were acquired every 10 min until worms reached adulthood. Growth, development, and fertility of worms were unaffected by these illumination conditions.

## Conventional image analysis of time-lapse imaging

With the exception of analysis shown in Figs 4E–4H and S9, worm segmentation and fluorescence measurements were conducted as previously described using a custom-made computational pipeline in Matlab [24]. In brief, the outline of the animals was detected using the Sobel algorithm by MATLAB edge() function applied to the fluorescence channel, followed by connecting the nearest endpoints to close gaps in the detected contours. Segmented worms were then straightened computationally, and the volume was inferred assuming rotational symmetry. Images where worm segmentation and/or straightening was faulty were detected by a decision-tree-based classifier based on shape features of the straightened animals. An ensemble of 20 bagged decision trees was trained on a subset of manually annotated images using MATLAB's TreeBagger() function. Images classified as faulty straightening or segmentation were excluded from further analysis. We validated that fluorescence intensity changes of up to 50% had no or negligible influence on segmentation quality and volume estimates (S11 Fig).

## Image analysis by neural network

For the measurements in Figs 4E, 4F, S8A, S8B, S9A, and S9B, segmentation was performed using a custom-trained deep neural network. We used an EfficientNetB5 encoder inside of a UnetPlusPlus architecture, implemented in Python (https://github.com/qubvel-org/segmentation_models.pytorch). For training, a dataset of 2000 images of worms expressing different ubiquitously expressed fluorescent markers and imaged using various microscopes, and at various stages of development (from eggs to adulthood) in agarose chambers were manually annotated with the help of MicroSAM [49]. 20% of the images were set aside for validation and 10% for testing. The model was trained for 1,000 epochs with an initial learning rate of $10^{-4}$ using the ADAM optimizer and an effective batch size of 24. We kept the model that performed the best on the validation set.

For calculation of growth rates, volume trajectories were median filtered with a window of 3 to remove extreme outliers missed by the error classifier and smoothed using Whittaker–Eilers smoothing [50] implemented in Python (https://github.com/AnBowell/whittaker-eilers/tree/main) with a smoothing strength of 0.75 and X being the time in hour since the start of the experiment each measurement was taken at. Confidence intervals for differences between populations were obtained by bootstrapping (1,000 repeats).

## Computation of volume and growth rates

The volume of individual worms was calculated at each time point from binary masks, assuming rotational symmetry. Larval stage transitions were identified by detecting the maximum of the second time derivative of the logarithm of animal volume, followed by manual correction using a MATLAB graphical user interface. Larval volumes at each molt (M1 to M4) were calculated using a linear regression of the volumes from the 10 time points before the molt. For the volume at birth, regression was based on the 10 time points after hatching. To compare growth rate and fluorescence trajectories of strain and conditions with different larval stage durations, each individual's trajectory was scaled by linearly interpolating each larval stage into 100 evenly spaced points, averaging across all individuals, and plotting as a function of larval stage progression [24]. Growth rates were calculated from the worm volumes, which were median-filtered with a three-time point window and further smoothed over 15 time points using MATLAB's smooth() function with the "rlowess" option. Growth rates were normalized to the duration of each larval stage, and individual signals were interpolated into 100 points per stage.

For comparison of the dynamics of growth rates and ribosomal protein concentrations, values were analyzed in individual animals using a 3.7-hour measurement window centered at 4.3 hours after hatching. Growth rates were computed by performing a robust linear regression on the natural logarithm of volume over this window, after applying a 3-point median filter. The slope of this regression represents the instantaneous relative growth rate ($d\log(V)/dt$). For ribosomal protein concentration, raw fluorescence intensities were background-subtracted, normalized to the animal's area, and then median-filtered using a 3-point window. The resulting values were normalized to the mean fluorescence of control conditions to obtain relative ribosomal protein concentrations. Data points where either growth rates or normalized ribosomal protein levels deviated by more than 3 standard deviations from their respective means were excluded as outliers.

## Quantification of germline lengths and fluorescence intensity

For quantification of RPL-34:mCherry expression in individual oocytes, oocytes were manually marked in the best focal plane centered on the nucleus. The nucleus was manually marked and excluded for intensity quantification. For the distal germline, the central plane of the distal region was manually identified and a region including the first 10 rows of nuclei was marked. For fluorescence quantification, manually drawn outlines were converted to segmentation masks using a custom-made ImageJ macro. Using a custom Matlab script, pixel values were summed up after background subtraction. And divided by the total area of each cell to obtain fluorescence values per pixel.

## Auxin-inducible degradation

For maternal treatments, auxin was added to the agar of standard NGM plates. For treatment in progeny, auxin was added to high- and low-melt agarose used to manufacture the chambers. Concentration was 500 µM except for depletion of RPS-26:AID:GFP, where 250 µM for *sun-1p*:Tir1 and 1,000 µM for *gld-1p*:Tir1 was used. For maternal auxin treatments, animals were exposed to auxin from hatching, except for DAF-2:AID, RPS-26:AID:GFP using the *eft-3p*:Tir1 and *sun-1p*:Tir1, where animals were grown to L4 stage on plates without auxin, and transferred to auxin containing plates 24h prior to collecting eggs for measuring progeny phenotypes.

## Polysome profiling

Polysome profiles were prepared as described in [51]. 100'000 L4 animals per strain were pulverized under cryogenic conditions in 20mM Tris-HCl (pH = 7.5), 100mM NaCl, 10mM $MgCl_2$, 1% Triton X100, 0.5mM DTT, 100 µg/ml CHX, 100U/ml RNase inhibitor (RNasin RNase inhibitor, Promega, N2515), and 10 µl/ml protease inhibitor cocktail (Sigma Aldrich, P8340) at 5cps in a SPEX 6750 Freezer/Mill (SPEX SamplePrep). Lysates were clarified by centrifugation at 4 °C, 3,000*g* for 3 min followed by an additional centrifugation step at 4 °C, 10,000*g* for 5 min. Ribosomes were resolved on 7%−47% sucrose density gradients in the prior described buffer for 3h at 35,000 rpm, 4 °C in a TH-641 rotor (Termo Scientific). Gradients were fractionated by upward displacement at 0.75 ml/min with continuous monitoring of $OD_{260}$ values and GFP fluorescence. mCherry fluorescence was measured in 50 µl per sampled fractions using a fluorescence plate reader (Agilent BioTek Synergy H1, Ex = 575 nm, Em = 620 nm).

## Supporting information

**S1 Fig. Global proteome changes in response to dietary restriction within and across generations. (A)** Quantification of maternal growth under DR and AL. Animals were sampled on days 3, 4, and 5 after cultivation in liquid medium with $2*10^8$ cfu/ml (DR) or $10^9$ cfu/ml (AL). Length (left) and number of eggs inside uterus (right) were measured for ≥5 individuals per condition. DR significantly reduces size and number of embryos in uterus (length: $p = 0.002$ for AL day 3 vs. DR day 4; $p = 0.02$ for AL day 3 vs. DR day 5; number of eggs in uterus: $p = 2.75*10^{-6}$ and $1.74*10^{-5}$, one-sided *t* test). No eggs were detected in DR animals after 4 days. Images show representative individuals at indicated sampling day. Scale bar = 100 µm. **(B)** Gene Ontology cellular component terms enriched among significantly downregulated (left) and upregulated (right) proteins in G0 adults under DR compared to AL conditions (FDR ≤ 0.05, minimum 2-fold change). Circle size indicates odds ratio and color intensity indicates significance, using an enrichment threshold of odds ratio > 3 and adjusted *p*-value < 0.01 (Benjamini-Hochberg correction). **(C)** Histograms comparing $\log_2$(fold change) in protein abundance between DR and AL conditions for G0 adults (blue) and G1 L1 larvae (orange). G0 adults show a broader distribution of fold changes (wider histogram) compared to G1 larvae, demonstrating larger proteome changes in the parental generation than in their progeny (mean absolute $\log_2$ fold change = 0.3 in G0 versus 0.11 in G1; variance = 0.078 in G0 versus 0.016 in G1). **(D)** Correlation between protein fold changes in G0 adults versus G1 L1 larvae ($R^2 = 0.037$), demonstrating weak inheritance of proteome changes across generations. **(E)** Principal component analysis (PCA) of protein abundance data from G0 adults, showing separation between AL and DR conditions across 3 replicates. **(F)** PCA of protein abundance data from G1 L1 larvae showing separation between progeny of AL and DR parents across 3 replicates. **(G)** Box plots showing $\log_2$(fold change) in abundance of ribosomal proteins (left) and proteasomal proteins (right) under DR conditions in G0 adults and G1 L1 larvae to respective AL conditions. *** indicates $p < 0.001$. **(H)** Images of animals carrying indicated fluorescent tags after maternal dietary restriction or *ad libitum* feeding followed by overnight starvation. Individuals closest to the respective population median are displayed. White color indicates highest expression. Scale bar = 20 µm. See S5 Data.
(TIF)

**S2 Fig. Ribosomal protein recovery in auxin-induced ribosome depleted progeny and in DR progeny.** Ribosomal protein levels relative to control during L1 development after ribosomal protein depletion in the maternal proximal germline

(dark blue), or after maternal DR (light blue). A value of 1 indicates full recovery of ribosomal protein levels compared to control animals. The recovery occurs faster after auxin-induced ribosomal protein depletion than after maternal DR. Solid lines: mean, shaded regions: 95% confidence interval. See S5 Data.
(TIF)

**S3 Fig. Phenotypes of maternal RPS-26:AID:GFP by *eft-3p*:Tir1 and *gld-1p*:Tir1. (A)** Growth rate during L1 development for *rps-26:aid:gfp, eft-3p:tir1* maternally auxin-treated progeny. **(B)** As (A), but RPS-26:AID:GFP fluorescence per pixel. **(C–E)** Fluorescence at hatch or ecdysis, volume at hatch or ecdysis, and larval stage duration. $n \geq 48$ from 2 days **(F–J)** As (A–E), but for *gld-1p:tir1.* central line: median, box: interquartile range (IQR), whisker: ranges except extreme outliers (>1.5*IQR), individual values: crosses, extreme outliers: circles. Total number of individuals $n \geq 36$ from 2 days, *** $p < 10^{-10}$ (Wilcoxon rank-sum test). Precise *p*-values and sample size in S2 Table. See S5 Data.
(TIF)

**S4 Fig. Insulin/Insulin-like Signaling (IIS) is not involved in intergenerational ribosomal protein control. (A)** Volume and RPL-34:mCherry concentration (pixel intensity normalized to control) at hatch of L1 progeny from mothers depleted for DAF-2:AID with 500 μM auxin from L4 stage onwards (yellow) and progeny of control animals (magenta). Progeny of DAF-2 depleted mothers are larger than progeny of control animals. This size increase is consistent with the increased size at hatch of *daf-2(e1370)* animals and after maternal somatic *daf-2* RNAi(14), validating effective DAF-2 depletion by AID. Maternal auxin does not reduce progeny RPL-34:mCherry concentration but slightly increases it ($p = 0.001$, Wilcoxon rank-sum test). central line: median, box: interquartile range (IQR), whisker: ranges except extreme outliers (>1.5*IQR), individual values: crosses, extreme outliers: circles. Number of individuals $n = 138$ and 141 from days. $p = 9*10^{-37}$ (Wilcoxon rank-sum test). **(B)** Growth rate of progeny during L1 development for parental depletion of DAF-2 and control. Individual trajectories were aligned at hatch point and M1 and re-scaled before averaging. Range between 100% and 120% represents the beginning of L2. Solid lines: mean, shaded regions: 95% confidence interval. Number of individuals and biological replicates as in (A). **(C)** As (B), but for RPL-34:mCherry concentration (fluorescence per pixel). **(D)** L1 larval stage duration, volume at hatching, and RPL-34:mCherry after maternal AL and DR in wild type (filled boxes) and *daf-16(mu86)* mutants (open boxes). *daf-16(mu86)* mutants are sensitive to maternal DR regarding their developmental rate and have altered volume at hatch, but these effects do not involve changes in the RPL-34:mCherry concentration. For each condition a total of at least $n \geq 100$ individuals were measured on at least 3 days. *** indicate $p < 10^{-5}$ (Wilcoxon rank-sum test). See S2 Table for precise sample size and p-values. **(E)** RPL-34:mCherry concentration as described for (C), but for indicated genotypes and maternal dietary treatments. Although *daf-16(mu86)* differ in the precise dynamics of RPL-34:mCherry recovery, the mutation does not affect the concentration immediately after hatching. See S5 Data.
(TIF)

**S5 Fig. Ubiquitous depletion of RAGA-1 by AID reduces ribosomal protein expression and delays growth. (A)** Larval stage duration of *raga-1:gfp:aid; eft-3p:tir-1; rpl-34:mCherry* strain treated with 500 μM auxin (gray) and control (red). central line: median, box: interquartile ranges (IQR), whisker: ranges except extreme outliers (>1.5*IQR), individual values: crosses, extreme outliers: circles. Number of individuals $n \geq 41$ from one day. **(B)** As (A), but for RPL-34:mCherry concentration (intensity per pixel normalized to control). *** indicate $p < 10^{-5}$ (Wilcoxon rank sum test). See S2 Table for precise sample size and *p*-values. See S5 Data.
(TIF)

**S6 Fig. RPL-34:mCherry concentration, volume at larval molts, and larval stage durations upon pharyngeal and ubiquitous maternal RAGA-1 AID. (A)** RPL-34:mCherry concentration (intensity per pixel normalized to control) of *raga-1:gfp:aid; eft-3p:tir1; rpl-34:mCherry* strain treated with 500 μM auxin (cyan) and control (magenta). central line: median, box: interquartile ranges (IQR), whisker: ranges except extreme outliers (>1.5*IQR), individual values: crosses,

extreme outliers: circles. **(B)** As (A), but for larval stage duration. **(C)** As (A), but for volume at larval molts. **(D, E)** As (A–C), but for *raga-1:gfp:aid; myo-2p:tir1; rpl-34:mCherry* strain (green) and control strain not expressing Tir1 (red), both treated maternally with 500 µM auxin. For each condition a total of at least $n ≥ 150$ individuals were measured on at least 2 days. *** indicate $p < 10^{-10}$ (Wilcoxon rank sum test). See S2 Table for precise sample size and p-values. See S5 Data. (TIF)

**S7 Fig. Phenotypic effects of maternal RAGA-1 depletion in the epidermis. (A)** Growth rate of progeny with and without depletion of maternal RAGA-1 in the epidermis. A strain expressing Tir1 under the *col-10* promoter was compared to a strain not expressing Tir1. Both strains were maternally treated with 500µM auxin. Individual trajectories were aligned at hatch point and M1 and re-scaled before averaging. Range between 100% and 120% represents the beginning of L2. Solid lines: mean, shaded regions: 95% confidence interval. **(B)** As (A), but for RPL-34:mCherry fluorescence (intensity per pixel normalized to control). **(C–E)** fluorescence at hatch/ecdysis, volume at hatch/ecdysis, and larval stage duration. central line: median, box: interquartile ranges (IQR), whisker: ranges except extreme outliers (>1.5*IQR), individual values: crosses, extreme outliers were omitted from display. For each condition a total of at least $n ≥ 99$ individuals were measured on 2 separate days. Delay in development of *col-10p*:Tir strain compared to control, occurring independent of RPL-34:mCherry changes, is likely due to leaky activity of Tir1 in the absence of auxin in this strain. See S5 Data. (TIF)

**S8 Fig. Maternal RAGA-1 depletion does not affect RPL-34:mCherry production during embryogenesis, but its levels in the oocytes. (A)** Concentration of RPL-34:mCherry in embryonic progeny of RAGA-1 depleted (dark, 500 µM auxin) and control (light, 0 µM auxin) mothers expressing Tir1 under the ubiquitous *eft-3p* promoter. Embryos were imaged at 10-minute intervals and individual trajectories were aligned to the time point of hatching ($t = 0$) prior to averaging. Shaded areas indicate 95% CI of the mean. n (control) = 87 individuals, *n* (maternal IAA) = 55 individuals, measured on 3 different days. **(B)** As (A), but for strain expressing Tir1 under the pharynx-specific *myo-2p* promoter. n (control) = 55 individuals, n (maternal IAA) = 46 individuals, measured on 3 different days. **(C)** RAGA-1-GFP-AID fluorescence intensity per pixel in the germline of RAGA-1 depleted mothers (red, 500 µM auxin) and control (blue, 0 µM auxin). Distal = most distal region of the germline (first 10 rows of nuclei), −4 to −1 = oocytes at indicated distance away from spermatheca. 1st embryo = egg positioned immediately after the spermatheca. Cytoplasmic fluorescence of oocytes was measured at the central focal plane of the nucleus, omitting signal from the nuclear area. Fluorescence is normalized to the median of the most highly expressed region of control conditions. RAGA-1 is significantly depleted in the entire germline ($p < 10^{-5}$ for all regions, one-sided rank sum test). central line: median, box: interquartile ranges (IQR), whisker: ranges except extreme outliers (>1.5*IQR), individual values: circles. $n > 15$ for all conditions and regions. **(D)** As (C), but for RPL-34:mCherry. RPL-34:mCherry is significantly depleted in all regions of the germline ($p$ from distal to 1st embryo: 0.002, 0.048, 0.002, 0.001, 2*$10^{-4}$, 2*$10^{-5}$, one-sided rank sum test). **(E)** As (D) but normalized to median of respective control for each region. Relative effect size of auxin treatment increases significantly from the distal end towards proximal regions (Spearman correlation coefficient = −0.34, $p$-value one-sided test = $10^{-4}$). **(F)** Same as (D), but for cell area. **(G)** Representative image of RAGA-1:GFP:AID fluorescence in control and auxin treated animals. Scale bar = 100µm. **(H)** As (G), but for RPL-34:mCherry fluorescence. See S2 Table for precise *p*-values and sample size, S5 Data. (TIF)

**S9 Fig. Dependence of intergenerational effect of dietary restriction on progeny growth conditions. (A)** Progeny of AL fed (blue) and DR mothers (orange) were grown with or without ubiquitous RAGA-1 AID (*eft-3p*:Tir1) in agarose chambers with abundant food. The difference in the relative (volume-specific) growth rate between progeny exposed to 500µM auxin or to vehicle control is plotted as a function of time after hatching. Shaded area indicates 95% CI. For each condition, a total of at least $n ≥ 49$ individuals were measured on 3 different days. **(B)** As (A), but for the growth rate disparity between DR and AL progeny under RAGA-1 AID and control conditions. **(C)** L2

duration of *eft-3p:tir1, raga-1-aid* animals with indicated maternal and progeny treatments. **(D)** As (C), but for total duration of development. **** *p*-value < $10^{-10}$, (Wilcoxon rank sum test). For precise *p*-values for each comparison, see S2 Table, S5 Data.
(TIF)

**S10 Fig. Validation of functionality of endogenously tagged ribosomal proteins. (A)** Quantification of larval stage durations of indicated strains in agarose chambers. Fluorescently tagged strains develop at near normal speed compared to a strain expression free GFP under control of the ubiquitous *eft-3* promoter, and similar to developmental durations measured on standard agarose plates. number of individuals $n$ = 13, 35, 29, 29, 40, 54, 45, 44, 40, 54, 45, 44, 40, 54, 39, 44 (from left to right) measured on 1 day **(B)** As (A), but for volumes at larval molts. Fluorescent ribosomal protein tagging causes a slightly decreased volume. **(C–E)** Polysome profiles of indicated strains as L4 animals feeding *ad libitum*. GFP fluorescence was measured together with RNA absorbance throughout the sucrose gradient. mCherry fluorescence was measured in collected fractions using a fluorescence plate reader. Data shows that all fluorescently tagged proteins localize to translationally active polysomal ribosomes. See S5 Data.
(TIF)

**S11 Fig. Dimming of fluorescence intensity does not significantly affect image segmentation and volume estimation. (A)** 100 images of animals were computationally dimmed by 50%. Dimmed and undimmed image sets were analyzed by the same algorithm using edge detection as described in Methods. No significant effect of fluorescence intensity on volume estimation is detected. **(B)** Jaccard Score of segmentation masks obtained from dimmed and undimmed images compared to manually curated ground truth. No significant decrease in quality is observed after dimming. **(C)** Jaccard Score of Segmentation performed on dimmed images, using segmentation masks of undimmed image as ground truth. Values near 1 (=perfect identify) show negligible effect of dimming on segmentation accuracy. See S5 Data.
(TIF)

**S1 Table. (A)** Significantly differentially regulated individual proteins after dietary restriction in adults with |$\text{Log}_2(FC)$| > 1 and at FDR < 0.05. **(B)** Significantly differentially regulated individual proteins after maternal DR with |$\text{Log}_2(FC)$| > 1 and at FDR < 0.05.
(XLSX)

**S2 Table. Number of individuals per experiment and statistics.**
(XLSX)

**S1 Data. Source data for Fig 1.**
(XLSX)

**S2 Data. Source data for Fig 2.**
(XLSX)

**S3 Data. Source data for Fig 3.**
(XLSX)

**S4 Data. Source data for Fig 4.**
(XLSX)

**S5 Data. Source data for Supplemental Figures (S1–S11 Figs).**
(XLSX)

## Acknowledgments

We are thankful to Cihan Elci and Ioana Gheorghe for technical assistance, Suzan Ruijtenberg (Utrecht University) for sharing strains prior to publication, and Frank Stein and Jennifer Schwarz (EMBL Heidelberg) for Proteomic analysis. We acknowledge support by the Microscopy Imaging Center at the University of Bern. We thank Jan Müller, Roman Ott, and Cristian Eggers for help with polysome profiling.

## Author contributions

**Conceptualization:** Sigma Pradhan, Benjamin D. Towbin.

**Data curation:** Sigma Pradhan, Sacha Psalmon, Benjamin D. Towbin.

**Formal analysis:** Sigma Pradhan, Sacha Psalmon, Joel Tuomaala, Benjamin D. Towbin.

**Funding acquisition:** Nicholas E. Stroustrup, Benjamin D. Towbin.

**Investigation:** Sigma Pradhan, Klement Stojanovski, Sacha Psalmon, Benjamin D. Towbin.

**Methodology:** Sigma Pradhan, Klement Stojanovski, Ferdinand Dellemann, Sacha Psalmon, Benjamin D. Towbin.

**Project administration:** Sigma Pradhan, Benjamin D. Towbin.

**Resources:** Nicholas E. Stroustrup, Benjamin D. Towbin.

**Software:** Sacha Psalmon, Benjamin D. Towbin.

**Supervision:** Benjamin D. Towbin.

**Validation:** Sigma Pradhan, Klement Stojanovski, Ferdinand Dellemann.

**Visualization:** Sigma Pradhan, Sacha Psalmon, Benjamin D. Towbin.

**Writing – original draft:** Sigma Pradhan, Benjamin D. Towbin.

**Writing – review & editing:** Sigma Pradhan, Benjamin D. Towbin.

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
