## [Editor Report · Decision Letter 0]

10 Mar 2025

Dear Dr Towbin,

Thank you for submitting your manuscript entitled "Intergenerational control of ribosomes under dietary restriction" for consideration as a Research Article by PLOS Biology. Please accept my sincere apologies for the delay in getting back to you as we consulted with an academic editor about your submission.

Your manuscript has now been evaluated by the PLOS Biology editorial staff, as well as by an academic editor with relevant expertise, and I am writing to let you know that we would like to send your submission out for external peer review.

IMPORTANT: After discussions within the editorial team, we would like to consider your manuscript as a Short Report (https://journals.plos.org/plosbiology/s/what-we-publish#loc-short-reports). Upon resubmission (details below), I would be grateful if you could please tick 'Short Report' as the article type.

Before we can send your manuscript to reviewers, we need you to complete your submission by providing the metadata that is required for full assessment. To this end, please login to Editorial Manager where you will find the paper in the 'Submissions Needing Revisions' folder on your homepage. Please click 'Revise Submission' from the Action Links and complete all additional questions in the submission questionnaire.

Once your full submission is complete, your paper will undergo a series of checks in preparation for peer review. After your manuscript has passed the checks it will be sent out for review. To provide the metadata for your submission, please Login to Editorial Manager (https://www.editorialmanager.com/pbiology) within two working days, i.e. by Mar 12 2025 11:59PM.

Kind regards,

Richard

Richard Hodge, PhD

rhodge@plos.org

PLOS

---

## [Decision Letter · Decision Letter 1]

7 May 2025

Dear Dr Towbin,

Thank you for your continued patience while your manuscript "Intergenerational control of ribosomes under dietary restriction" was peer-reviewed at PLOS Biology. Please accept my sincere apologies for the delays that you have experienced during the peer review process. Your manuscript has been evaluated by the PLOS Biology editors, an Academic Editor with relevant expertise, and by three independent reviewers.

As you will see in the reviewer reports, which can be found at the end of this email, although the reviewers find the work potentially interesting, they have also raised a substantial number of important concerns. Based on their specific comments and following discussion with the Academic Editor, it is clear that a substantial amount of work would be required to meet the criteria for publication in PLOS Biology. However, given our and the reviewer interest in your study, we would be open to inviting a comprehensive revision of the study that thoroughly addresses the reviewers' comments and convincingly shows that the germline is not the primary driver of the observations, but rather somatic mTORC1 signaling affects the germline. Given the extent of revision that would be needed, we cannot make a decision about publication until we have seen the revised manuscript and your response to the reviewers' comments. Your revised manuscript would need to be seen by the reviewers again, but please note that we would not engage them unless their main concerns have been addressed.

The reviewers generally think the findings are interesting and that the manuscript is well done. Reviewer #1 is broadly positive and raises two conceptual issues, as well as requesting several clarifications and textual edits. After discussions with the Academic Editor, we ask that the two conceptual issues raised by Reviewer #1 are addressed experimentally and we consider this essential for the revision. Reviewer #2 notes that further validation for the role that mTORC1 activity plays in modulating progeny ribosome content is required. In addition, he/she raises concerns that ribosome concentration measurements rely on tagged ribosomal proteins, so further functional validation is needed since fluorescence intensity may not fully reflect functional ribosome concentration. Finally, Reviewer #3 is critical about the experimental design since both the parents and progeny are starved, so the assay is not measuring intergenerational protein levels. Please note that the Academic Editor is less concerned with this point about F1 starvation because this is present in the control as well. Therefore, we will make experiments to directly address this optional, since there is an intergenerational effect of adult P0 starvation demonstrated due to a difference between F1s from starved and non-starved P0.

We appreciate that these requests represent a great deal of extra work, and we are willing to relax our standard revision time to allow you 6 months to revise your study. Please email us (plosbiology@plos.org) if you have any questions or concerns, or envision needing a (short) extension.

**IMPORTANT - SUBMITTING YOUR REVISION**

*Resubmission Checklist*

*Published Peer Review*

*PLOS Data Policy*

*Blot and Gel Data Policy*

Best regards,

Richard

Richard Hodge, PhD

rhodge@plos.org

REVIEWS:

Reviewer #1: The manuscript describes the intergenerational effect of dietary restiction on progeny growth and discovers that an observed downregulation of ribosomes in the progeny is largely sufficient for recapitulating this early growth impairment. The authors find that depletion of maternal somatic ribosomes is sufficient to induce a depletion in the progeny, and that maternal somatic depletion of the mTORc1 subunit RAGA-1 was also sufficient to induce an intergenerational downregulation of progeny ribosomes.

This manuscript is clear and well-written and represents a carefully executed body of work which definitely merits publication. It has likely relevance outside of the model organism in which the work was carried out and serves as an important example of the ways in which broad material transfers from mothers to their progeny can underlie intergenerational phenotypes, although they are often overlooked in favour of more obviously 'information-encoding' modes of inheritance such as small RNAs, as the authors highlight in the Discussion.

That said, there are a couple of broad conceptual issues which the authors might be able to address either experimentally or by further discussion, in addition to some formatting and/or typological errors or ambiguities.

CONCEPTUAL:

1) The authors speculate in the Dicussion in lines 381-387 and in their concluding lines (406-407) about the likelihood that this ribosomal downregulation may be adaptive or at least not costly for the progeny under expected conditions:

381-387: 'such a reduction is not expected to negatively impact progeny growth and fitness if dietary restriction continues across generations.'

406:407: 'Our work highlights the allocation of proteome resources to ribosomes as a new mechanism for the intergenerational transmission of phenotypic plasticity that may prepare progeny for environmental conditions that they are most likely to encounter.'

It strikes me that the experimental systems that the authors describe should be able to address these conjectures. In particular it should be possible to quantify the growth disparity of DR vs AL progeny under DR conditions, not only under AL conditions. If 'such a reduction is not expected to negatively impact progeny growth and fitness if dietary restriction continues across generations', then that predicts that this growth disparity should no longer be evident or should be negligible under continued DR conditions. Can this be tested?

2) The authors suppose in the Discussion that the inherited 'signal' may be ribosomes deposited in the oocyte that were synthesised in the proximal germline. They show that ribosome depletion in the maternal soma and the distal germline do not recapitulate the intergenerational growth phenotype but they do not perturb ribogenesis in the proximal gonad or directly examine the response of ribogenesis in the proximal gonad to somatic RAGA-1 disruption. It would be nice to have some more evidence that the 'signal' involves ribogenesis specifically in the proximal gonad.

TYPOGRAPHICAL ISSUES

- Several places compound adjectives should be hyphenated. e.g.

37 near-linear

120 membrane-bound

311 somatically-reduced

374 long-term

- line 43 under conditionS of DR

- line 73 'progeny of DR animals WERE exposed'

- line 93 'Downregulated proteins were enriched for translational machinery, particularly in ribosomes^ and vitellogenins (yolk proteins) while upregulated proteins were...'

Comma needed at ^? Yolk proteins are not translational machinery

- lines 89, 448. Numbers to be written as 1,000 not 1'000

- Fig S1B: typo 'Upwnregulated'

AMBIGUITY

line 86 - ambigious phrasing 'Dietary restriction (DR) was imposed by a five-fold food dilution compared to ad libitum (AL) feeding (2*108 vs. 109 cfu/ml), which slowed down development from three to five days per generation (Fig. 1A).'. Although I understand the sense of it and it is clear with reference to Fig 1A, it could be read as that DR slowed down development 3-5 days per generation, which isn't what they mean. Rephrase to avoid any potential confusion.

Fig S1A is a strange figure that contains very little information; in the end the information contained could easily be reported in the text in two figures (6454 proteins for P0, 6432 for F1). Also it seems odd that the number of proteins detected is exactly the same for each sample within each generation (but slightly different across generations). Is this correct?

line 386; use plain English in the Discussion when referring to the DR to AL shift. Here to make the broader point it is better to avoid the jargon and abbreviations.

TERMINOLOGY

line 54: 'called intergenerational inheritance when lasting for only one generation, or transgenerational inheritance when lasting for longer'. While the point is not very relevant to the current paper this isn't always correct for all organisms. See Heard & Martienssen 2014 [PMID 24679529] Fig 1 for nuance. Some would argue even in C. elegans that a 2 generation effect might be considered intergenerational (as the germ cells may already be present in progeny in utero when their grandparents are exposed to an environmental stimulus during adulthood). 'Parental effects' would be limited more specifically to one generation.

At the risk of being pedantic, F1 stand for Filial generation and is derived from a genetic cross. As that is not the case here, P0 / F1 might be better expressed as G (for generation) 0 / G1

lines 160/209/206/270: the authors several times use m e.g. m = 3. As this is repeated I assume this is deliberate, I guess to distinguish from the reported n (which does stand for 'number' of individuals). Nonetheless this convention of using m was unfamiliar and unintuitive to me, as it is not obvious what 'm' stands for.

Reviewer #2: In this study, the authors combine proteomics, genetics, and live imaging in C. elegans to uncover an intergenerational regulatory mechanism in which maternal dietary restriction leads to reduced ribosomal protein levels in offspring. While most components of the proteome are reset between generations, ribosomal proteins remain reduced in progeny, delaying their growth until ribosome levels normalize. The authors show that this effect is mediated by mTORC1 signaling in the maternal soma, as soma-specific depletion of RAGA-1 mimics dietary restriction. These findings suggest that maternal mTORC1 activity communicates across the soma-germline barrier to program ribosomal provisioning in offspring, revealing a novel axis of intergenerational regulation.

This study presents a compelling conceptual advance and will be of interest to a wide audience. The data is generally well-analyzed. However, there are major points that should be addressed to support the conclusions made.

Major Comments:

Somatic specificity of mTORC1 depletion via eft-3p::TIR1 requires further validation.

The central conclusion that maternal somatic mTORC1 activity modulates progeny ribosome content hinges on the assumption that eft-3p::TIR1-driven RAGA-1 degradation is restricted to somatic tissues. While eft-3 is typically used as a pan-somatic promoter, it is derived from a translation elongation factor and has been reported to drive detectable expression in the germline under some conditions—especially over long auxin exposures or high concentrations (e.g., 500 μM to 1 mM for >18 hours). In this study, RAGA-1::GFP depletion was assessed after 26 hours of 500 μM auxin, and while soma-specific reduction is visually presented (Fig. 4A), quantification of RAGA-1::GFP signal in the germline versus soma would be essential to convincingly demonstrate specificity. This could be done either in whole animals or dissected gonads, using the same imaging conditions.

Alternatively (or additionally), the authors could support this claim by using a more germline-restrictive strategy, either by repeating the experiment with a validated somatic-only TIR1 driver such as rgef-1, myo-3, dpy-7, or col-10 (as described in Ashley et al., Genetics 2021, PMID: 33677541), or by employing a mosaic approach as described in Artiles et al. (Dev Cell, 2019, PMID: 30799227), which allows for tissue segregation in F1. While the mosaic analysis is technically more demanding, even a limited cohort of 5-10 F1s with clear soma vs germline separation would significantly strengthen the conclusion that the effect is truly somatic in origin.

Ribosome concentration measurements rely on tagged RPs, here functional validation is needed. The conclusion that ribosomal protein levels are reduced in progeny is primarily based on fluorescent intensity measurements of RPL-34::mCherry and RPL-29::GFP. However, these are transgene insertions, and it remains unclear whether the tagged proteins are functionally incorporated into mature ribosomes or able to complement loss of function alleles. Without such validation, fluorescence intensity may not fully reflect functional ribosome concentration but instead also detect free unassembled ribosomal proteins. At minimum, the authors should clarify in the Methods whether the alleles are functional (e.g., rescue data, absence of growth defects), and whether incorporation into ribosomes has been previously demonstrated ( polysome analysis, or prior publications). If this information is not available, the conclusions regarding "ribosome concentration" should be stated more conservatively as "fluorescence levels of tagged ribosomal proteins."

In Figure 3, degradation of RPS-26 in the germline (using sun-1p::TIR1) results in reduced progeny growth and ribosome concentration, whereas depletion of RPL-22 in the soma (eft-3p::TIR1) or distal germline (gld-1p::TIR1) does not. However, no tissue matched depletion of the same RP is shown so it's unclear whether the difference is due to tissue specificity or intrinsic differences between RPs.

What is the correlation between dietary restriction (DR) and the mTORC1 pathway in the intergenerational regulation of ribosomes and growth rate? Does dietary restriction act through mTORC1? Depletion of RAGA-1 in the maternal soma(eft-3p:TIR1) phenocopies the effects of maternal dietary restriction. What are the phenotypes resulting from RAGA-1 depletion in the proximal or distal germline? Finally, could mTORC1 activation during DR rescue or suppress ribosome concentrations and growth rate? Evidence or elaboration on this would greatly improve the manuscript.

Minor Comments:

Line 89: Change "6'400 proteins" to "6,400 proteins" for formatting consistency.

Line 103: Define the KS test acronym before first use.

Line 113 / Figure 1D: Please clarify whether GO enrichment analysis was based on significantly regulated proteins or on z-scored trends across generations. The source and method for the GO analysis should also be described ( tool or database used).

Line 456: Consider updating the section header to "TMT Mass Spectrometry and Analysis" for clarity.

Methods section: Please include full details of auxin treatment (concentration, duration, temperature, developmental stage, etc.) for each relevant experiment.

It appears that Table S1b is missing, and the list of differentially expressed genes shown in Figure 1E has not been provided.

Reviewer #3: In the manuscript "Intergenerational control of ribosomes under dietary restriction" Pradhan et al demonstrated that dietary restrictions (DR) in the parental generation lead to phenotypic changes in the offspring (F1), characterized by a slower growth rate and reduced body size during early development, specifically in the larval stage (L1). The authors then performed quantitative proteomics and P0 and then double starved F1s to reveal that ribosomal proteins were significantly reduced in both the parental generation (P0) at the young adult stage and in the F1 reexposed offspring at the L1 stage, which contributes to these phenotypes (slow growth and decreased body size). The authors also depleted certain ribosomal proteins in the P0 germline, which resulted in a phenotype similar to that of the DR mothers. Additionally, the study demonstrates that somatic RAGA-1 signaling is crucial in the parental generation, as it regulates ribosomal protein levels in the next generation (F1). Depleting somatic RAGA-1 in the P0 was sufficient to replicate the DR phenotype in F1.

The authors phrasing of this manuscript has substantial problems because they have starved the F1 worms to achieve the arrested L1 stage which means that they are starving both the parents and the children, and this is not really measuring intergenerational protein levels but whether repeated starvation is different from single starvation. This is still an interesting question but substantially different from how the authors introduce the topic, therefore the introduction, title, and conclusions need to be rewritten to reflect what was performed. Additionally, reduced levels of maternally loaded ribosomal proteins or other vitellogenin proteins (yolk proteins) in the mothers due to DR could be the cause of the observed phenotypic changes. It is possible that DR does not directly alter ribosomal protein expression in F1, but rather results in decreased maternal ribosome loading, which leads to minor phenotypic changes in early L1 progeny. Furthermore, a previous study (PMID: 30799226) indicates that maternally loaded ribosomes are critical for early development and are sufficient for the F1 progeny to produce fully functional first-stage larvae, with no change in body length at hatch observed even when ribosomal proteins in F1 are non-functional.

In addition to these below are some experimental suggestions that will significantly improve the manuscript.

Major points

1) Arrested L1 larvae is achieved how? This seems to be achieved by bleaching and then growing the eggs in media without food which therefore eliminates anything inherited as you are subjecting it to a second successive generation of starvation. This is a very different question than the way you have introduced the paper. The intro and the conclusions therefore need to be rewritten to reflect what you have actually done. When you try to make conclusions about comparing the P0 to the F1 this is obviously impossible to do as you are comparing apples to oranges.

2) It would strengthen the paper to perform your auxin treatment of RPS-26 and raga-1 with or without parental DR and see if the ribosome depletion or TOR depletion further exacerbates the developmental slowing or if they are truly epistatic. Otherwise, you have two separate findings, that parental starvation + additional kid starvation leads to reduced ribosomes and that that same treatment leads to reduced growth rates.

3) Furthermore does mTOR/RAGA-1 depletion in somatic tissues also influence ribosome content in the oocyte? Does it affect the expression of other vitellogenin family proteins in the mother's intestine? Since these proteins are ultimately transferred to the oocyte, they play a critical role in the early development of the progeny. How the RAGA-1 in soma of P0 regulate the expression of ribosome in the F1?

4) It is important to analyze the oocyte proteome following mTOR/RAGA-1 depletion in the soma to check the levels of maternally loaded proteins.

5) The authors should assess the levels of maternally derived proteins under DR and AL conditions in order to substantiate their claim that maternal diet regulates ribosomal protein expression levels in F1, rather than attributing the changes to reduced maternal ribosome loading.

Minor points

1) Are there other intergenerational phenotypes in response to parental DR? Figure 2D needs to be shown as Figure 1A.

2) Where is your DR protocol coming from? Is this something you created out of thin air or a published protocol? If published need to cite, if created from nothing need to fully characterize the protocol and its phenotypes. 1A does not have any results but is merely a model. If you are stating that there is a developmental slowing you need to quantify and demonstrate it is statistically significant.

3) It would strengthen figure 1 F to show representative images so the readers can see what that fold change actually looks like.

4) I am unclear as to how you use a protein, RPL-34, which is downregulated in response to DR as your marker for Figure 2. Don't you need to use a protein that isn't changing overall level? Again 2B needs a representative image so the readers can see if an 11% change is actually visible. Might be good to complement with a western blot so that the IF can be a little more quantitative.

5) I think it would be important to show polysome profiles for your different conditions. Do these look the same or different?

6) All graphs need to start at 0. You can add dashed lines in the axis if you want to focus on a particular region but if it doesn't actually start at 0 you are misleading the reader about the magnitude of these changes.

7) The mass spectrometry (MS) data shows that members of the vitellogenin (vit) family of proteins are among those most significantly downregulated under DR conditions in mothers. These proteins are maternally loaded into the oocyte and are crucial for early F1 growth and development. Therefore, it is important to investigate whether this effect is influenced by oocyte receptor mutants, such as RME-2(-), which help transport vit proteins into the oocyte. This would confirm that the observed phenotypes in F1 are not a result of lowered maternal vit proteins.

8) Figure 3 (A-E) clearly shows that depleting maternal rps-26 in the germline (P0) also reduces its level in the progeny (F1), suggesting that the phenotypic changes are due to lower levels of maternally inherited proteins rather than changes in their expression.

9) In Figure 3 (F-H), rpl-22:aid; eft-3p:tir1 was treated with auxin, which degrades rpl-22. However, the authors also checked the level of rpl-34:mCherry. The rationale for this comparison is unclear to me. Why does the level of rpl-34:mCherry decrease following auxin treatment? This data further suggests that somatic ribosomal levels in the parent do not affect progeny development; rather, it is the germline ribosomal proteins that are significant.

---

## [Decision Letter · Decision Letter 2]

10 Feb 2026

Dear Dr Towbin,

Thank you for your patience while we considered your revised manuscript "Intergenerational control of ribosomes under dietary restriction" for publication as a Short Report at PLOS Biology. Please accept my sincere apologies for the delays that you have experienced during this round of the peer review process. This revised version of your manuscript has been evaluated by the PLOS Biology editors, the Academic Editor and two of the original reviewers.

Based on the reviews, I am pleased to say that we are likely to accept this manuscript for publication, provided you address the following data and other policy-related requests that I have provided below (A-E):

(A) We routinely suggest changes to titles to ensure maximum accessibility for a broad, non-specialist readership. In this case, we would suggest a minor edit to the title, as follows. Please ensure you change both the manuscript file and the online submission system, as they need to match for final acceptance:

“Dietary restriction shapes intergenerational ribosome abundance and early growth of C. elegans offspring”

(B) You may be aware of the PLOS Data Policy, which requires that all data be made available without restriction: http://journals.plos.org/plosbiology/s/data-availability. For more information, please also see this editorial: http://dx.doi.org/10.1371/journal.pbio.1001797

-Supplementary files (e.g., excel). Please ensure that all data files are uploaded as 'Supporting Information' and are invariably referred to (in the manuscript, figure legends, and the Description field when uploading your files) using the following format verbatim: S1 Data, S2 Data, etc. Multiple panels of a single or even several figures can be included as multiple sheets in one excel file that is saved using exactly the following convention: S1_Data.xlsx (using an underscore).

-Deposition in a publicly available repository. Please also provide the accession code or a reviewer link so that we may view your data before publication.

Figure 1B-F, 2B-G, 3B-E, 4A-H, S1A, S1C-G, S2, S3A-J, S4A-E, S5A-B, S6A-F, S7A-E, S8A-F, S9A-D, S10A-E, S11A-C

(C) Further to point (B), I note that you have already provided the underlying data in the Source Data file. I have checked and this looks good, but it seems that data for Figures 4E-H and S11A-C are missing from this file?

(D) Thank you for providing the mass spectrometry proteomic data in the ProteomeXchange (PXD060999). However, I searched for the deposition and was unable to find it. I would be grateful if you could check whether the accession number is correct or whether the data has been made publicly available at this stage.

(E) Please also ensure that each of the relevant figure legends in your manuscript include information on *WHERE THE UNDERLYING DATA CAN BE FOUND*, and ensure your supplemental data file/s has a legend.

We expect to receive your revised manuscript within two weeks.

*Published Peer Review History*

*Press*

Best regards,

Richard

Richard Hodge, PhD

rhodge@plos.org

Reviewer #1 (Marcos Francisco Perez, identifies himself): The authors have done a fine job of addressing my comments and those of the other reviewers. I am satisfied that the manuscript is fit for publication.

Reviewer #2: The authors have generally responded to my comments and have made substantial revisions to avoid overstating their conclusions. They revised their language regarding soma-specific degradation to refer instead to maternal depletion.

The authors also added new experiments using tissue restricted manipulations to strengthen their conclusions about somatic contributions. Pharynx-specific RAGA-1 depletion recapitulates key aspects of the intergenerational phenotype, whereas epidermal-specific depletion affects maternal growth without altering progeny ribosomal protein levels. These experiments provide stronger support for a soma-driven mechanism rather than a trivial consequence of reduced maternal growth.

The authors also addressed concerns regarding ribosomal protein measurements by adding functional validation and by refining their terminology throughout the text, which improves clarity. The inclusion of raga-1 gain-of-function data is a valuable addition: hyperactivation of mTORC1 partially suppresses the L1 duration, supporting the idea that mTORC1 activity is connected with the growth phenotypes.

Overall, the revisions substantially strengthen the manuscript, and the added data address the major concerns raised in the initial review.

---

## [Editor Report · Decision Letter 3]

23 Feb 2026

Dear Dr Towbin,

On behalf of my colleagues and the Academic Editor, Peter Sarkies, I am pleased to say that we can accept your manuscript for publication, provided you address any remaining formatting and reporting issues. These will be detailed in an email you should receive within 2-3 business days from our colleagues in the journal operations team; no action is required from you until then. Please note that we will not be able to formally accept your manuscript and schedule it for publication until you have completed any requested changes.

PRESS

Best wishes,

Richard

Richard Hodge, PhD

rhodge@plos.org

PLOS
